# The non-catalytic DNA polymerase ε subunit is an NPF motif recognition protein

Salla Keskitalo [1,6], Boglarka Zambo [2,3,6], Dicle Malaymar Pinar [1], Antti Tuhkala [1], Kari Salokas [1], Tanja Turunen[1], Norbert Deutsch [4], Norman Davey [5], Zsuzsanna Dosztányi [4], Markku Varjosalo [1] ✉ & Gergo Gogl [2,3] ✉

Short linear motifs (SLiMs) in disordered protein regions direct numerous protein–protein interactions, yet most remain uncharacterized. The Asn-Pro-Phe (NPF) motif is a well-known EH-domain ligand implicated in endocytosis, but here we reveal that the non-catalytic subunit of human DNA polymerase ε (POLE2) also serves as a general NPF-motif receptor. Using a quantitative "native holdup" assay, we find that POLE2 selectively binds diverse NPF-containing peptides, including canonical EH-domain ligands (e.g., SYNJ1) and previously uncharacterized motifs. Biochemical measurements and mutational analyses show that NPF motifs interact with a shallow pocket near the POLE2 C-terminus, and AlphaFold predictions confirm key roles for Y513, E520, and S522 in motif coordination. Proteome-scale affinity screens identify NPF-containing nuclear proteins (e.g., WDHD1, DONSON, TTF2) that bind POLE2 with micromolar affinities, and their motif mutations abolish binding in cell extracts. Although POLE2 primarily tethers the catalytic POLE subunit to replication forks, these results indicate that it can also recruit various NPF-bearing partners involved in replication, DNA repair, and transcription regulation. Notably, NPF motifs optimized for EH-domain binding can still associate with POLE2, highlighting the inherent degeneracy of SLiM-mediated networks. Overall, these findings establish POLE2 as a central hub possibly linking replication with other processes via broad NPF-motif recognition.

Regions showing strong local evolutionary conservation frequently occur in otherwise poorly conserved, intrinsically disordered protein regions. Many of these mediate protein-protein interactions and these sites are often referred to as SLiMs (**s**hort **l**inear **i**nteraction **m**otifs) or simply as motifs[1,2]. Motifs are indispensable for complex biological processes and constitute the majority of our cellular interactome generating a great interest in identifying novel motifs and motif recognition proteins. To this end, several computational de novo motif discovery tools have been developed to identify putative motifs in disordered protein regions based on their unique amino acid compositions and evolutionary properties[3,4]. Although these approaches identify numerous putative motifs, they cannot predict their binding partners, leaving them as orphan motifs. Finding their unknown partners poses significant experimental challenges, since motif-mediated interactions typically have weak affinities and fast binding kinetics[5]. These properties make them practically invisible for mainstream

[1]Institute of Biotechnology, Helsinki Institute of Life Science HiLIFE, University of Helsinki, Helsinki, Finland. [2]Institut de Génétique et de Biologie Moléculaire et Cellulaire (IGBMC), INSERM U1258/CNRS UMR 7104/Université de Strasbourg, Illkirch, France. [3]Institut de Biologie Valrose – iBV, Université Côte d'Azur, CNRS UMR7277, Inserm, Nice, France. [4]Department of Biochemistry, ELTE Eötvös Loránd University, Budapest, Hungary. [5]Institute of Cancer Research, Chester Beatty Laboratories, London, UK. [6]These authors contributed equally: Salla Keskitalo, Boglarka Zambo. ✉e-mail: markku.varjosalo@helsinki.fi; gergo.gogl@univ-cotedazur.fr

interactomic approaches. Fortunately, it is possible to exploit the degenerate nature of motif-mediated interactions to predict the putative binding partner of a given orphan motif in certain cases. By comparing the sequence of an orphan motif with catalogs of consensus sequences of known motif-recognizing proteins, it is sometimes possible to assign the putative motif to a given interaction partner[6]. However, a simple consensus motif can be often recognized by multiple members of an entire protein family[7], or members of multiple protein families[8] and simplistic consensus motifs do not contain sufficient information to distinguish between interactions of different strength[9]. Adding to these limitations, we are currently unaware of how many motif recognizing proteins may be present in our proteome and only a few hundred consensus motifs have been cataloged so far[10].

The case of NPF (Asn-Pro-Phe) motifs illustrates these critical issues. Almost three decades ago, it was discovered that NPF motifs can bind to EPS15 homology (EH) domains[11–13] (Supplementary Fig. 1A). EH domains are calcium-binding EF-hand domains found in eleven human EH proteins[14] (Supplementary Fig. 1B). Multivalency is prevalent for EH proteins as two of them contain three EH domains and four of them contain two domains[14]. Moreover, even the four EHD proteins that contain only one EH domain form stable oligomers[15]. The partners of these proteins contain disordered protein regions containing NPF motifs, often present in multiple copies, contributing to multivalent and synergistic interactions[16]. EH proteins primarily function during vesicle formation, and their NPF motif-mediated interactions are crucial for protein sorting[16,17]. These interactions exploit the stable Asx turn conformation formed by the NPF tripeptide, where the carbonyl of the Asn sidechain forms a hydrogen bond with the amide nitrogen of Phe residue of the NPF motif[13]. Since all members of the EH family recognize the same structural motif, a putative NPF motif can interact with any member of the family. In addition, many additional proteins have been reported to be able to bind NPF motifs, that are unrelated to EH domains (Fig. 1A). NPF motifs are also involved in vesicle budding interactions of SNAREs through mediating interactions with the SEC23/24 subunits of COPII[18,19]. The NPF motif of Pygo2 can be recruited at the composite interface formed between an SSBP2 dimer and LDB1 and their ternary complex forms the so called WNT enhanceosome[20,21]. An N-terminal NPF motif in BORA was found to be part of a critical region for activating Aurora A kinase (AURKA), and it was proposed that this motif may interact with the AURKA kinase domain[22]. A fungal protein called SHD1 binds to NPF motifs that functions as endocytic internalization motifs[23]. In a phage display experiment, the HIV-1 envelope glycoprotein gp41 was found to have a strong preference for binding HxxNPF motifs and such peptides were able to inhibit HIV-1 envelope glycoprotein-mediated syncytium-formation[24]. Finally, four studies were published in parallel to our investigation, two reporting that the non-catalytic ε DNA polymerase subunit (POLE2) can interact with NPF motifs found in Protein downstream neighbor of Son (DONSON)[25,26], and two describing an interaction between POLE2 and the transcription termination factor 2 (TTF2)[27,28]. Besides these NPF binding proteins, our proteome contains many additional Pro-rich motif recognizing proteins – like SH3 domains that bind to PxxP motifs or PTB domains that bind to NPxY motifs – and their binding motifs may also partially overlap with NPF motifs[29]. As illustrated above, a putative NPF-containing motif can potentially bind to large numbers of partners, including the 11 type EH proteins, COPII, SSBP2/LDB1, AURKA, POLE2, and possibly many others including proteins from pathogens. Currently, it is not yet explored what are the determinants of specificity between these different types of interactions and it is also unknown how widespread are the interactomes of each of these NPF recognition proteins.

Here, we developed an experimental pipeline to identify partners of NPF motifs and found that POLE2 functions as a general NPF motif recognition protein. We unraveled a proteome-wide NPF motif-mediated interaction network of POLE2, we investigated their binding mechanism and determined the key specificity determinants of NPF motifs between POLE2 and an EH protein, called EPS15. Our findings provide new insights into the broad interaction network of DNA polymerase ε and its role in the replisome. Furthermore, this study highlights the fundamental specificity issue in motif-mediated interactions and presents a comprehensive, unbiased experimental framework to address it.

## Results

### Identification of POLE/POLE2 as a recurring interaction partner of unrelated NPF motif-containing peptides

To identify the partners of NPF motifs, we used the native holdup (nHU) assay that was recently developed by our team (Fig. 1B). The holdup approach has been extensively used to determine equilibrium dissociation constants of transient interactions mediated by PDZ and SH3 domains, 14-3-3, and human papilloma virus E6 proteins, and even nucleic acid-protein interactions[7,30–35]. In the nHU experiments, a purified bait molecule, or a control compound, is immobilized on a resin at high concentration (typically 10 μM, or higher) and these resin stocks are subsequently mixed with dilute cell extracts. After a relatively long incubation, i.e., 2 h at 4 °C, the liquid phase is separated by centrifugation. This liquid phase contains the unbound prey molecules at near binding equilibrium that can be quantified using selective and quantitative analytical approaches, such as mass spectrometry. Therefore, proteins that interact with the bait molecule will appear to be depleted in the solvent phase and by measuring its relative concentration compared to the control nHU experiment, the degree of binding ($\theta$) can be determined that can be converted to apparent steady-state dissociation constants, according to:

$$\theta = 1 - \frac{I_{\text{free}}}{I_{\text{tot}}} = \frac{C_{\text{bait}}}{C_{\text{bait}} + K_{\text{app}}} \tag{1}$$

where $I_{\text{tot}}$ and $I_{\text{free}}$ are the measured total and bait-depleted equilibrium mass spectrometry total ion intensities of prey proteins, $C_{\text{bait}}$ is the bait molecule concentration and $K_{\text{app}}$ is the apparent dissociation constant.

To demonstrate that nHU could be used to characterize interactions of NPF motifs, we selected three previously reported NPF motifs from SYNJ1, BORA and PYGO2, that are expected to bind to EH proteins, AURKA and SSBP2/LDB1, respectively. We also included three orphan or poorly studied NPF motifs from DONSON (likely to bind to POLE2), WDHD1, and WEE1 that were predicted by the SLiMPrints approach over a decade ago[4]. We synthesized these motifs as biotinylated peptides to use them in nHU experiments as baits using biotin in the control experiment. Subsequently to nHU assay, we analyzed the depleted extracts with label free quantitative mass spectrometry (MS) and identified statistically significant interactions as previously described[35] (Fig. 1C–H, Supplementary Data 1). As expected, the SYNJ1 NPF-motif rich region – containing three NPF motifs – showed significant interaction with EH proteins EHD1, EPS15, EP15R, ITSN2 and showed substantial depletion, albeit below our stringent significance threshold in the EH proteins SYNRG and REPS1. The NPF motif of WEE1 depleted EHD1, albeit the statistical significance of this interaction was below our strict threshold. The motif of BORA also depleted EHD1, as well as AURKA. PYGO2 was found to interact with both EHD1, EHD4, and AURKA, besides the core WNT enhanceosome, consisting of LDB1 and SSBP2/4. The NPF motif of DONSON showed significant interaction with two SH3 domain-containing proteins, CD2AP and SH3KBP1. Most surprisingly, the NPF motif peptides of both DONSON, WDHD1 and even SYNJ1 showed significant binding to the heterodimer of catalytic and the non-catalytic subunits of the ε DNA polymerase, called POLE and POLE2, respectively. Although the POLE/POLE2 complex was already suspected to interact with DONSON through its NPF

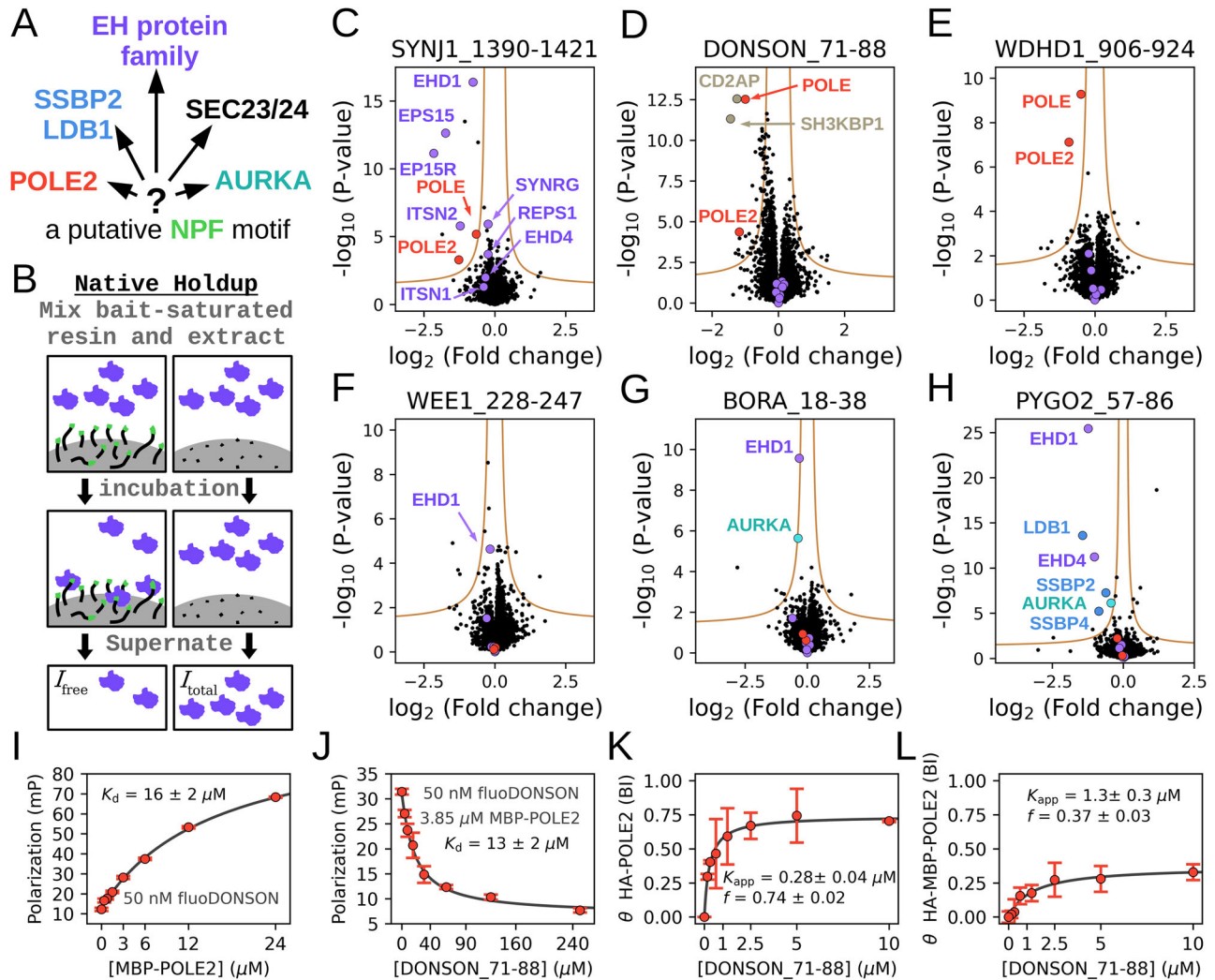

**Fig. 1 | NPF motifs bind to various types of proteins, including POLE and POLE2.**
**A** NPF tripeptides were described to interact with multiple types of NPF motif receptors, making prediction of partners of putative motifs highly ambiguous. **B** The native holdup (nHU) affinity interactomic approach is ideal to capture motif-mediated interactions because of high ligand concentration allowing abundant complex formation of weak interactions and no washing procedures that would eliminate transient interactions. **C–H** Six different NPF motif peptides were assayed with nHU coupled with mass spectrometry. Interaction partners belonging to known or suspected NPF motif receptors are highlighted. Experiments were carried out with 2 biological and 3 technical replicates (total $N = 6$). Two-tailed unpaired T-test, assuming equal variance was used during analysis. POLE/POLE2 (in red) and detected EH proteins (in violet) are shown on the volcano plots even if they were not identified as partners to show specificity of NPF motifs. **I** Fluorescence

polarization was measured by titrating recombinant avi-His$_6$-MBP-POLE2 to a fluorescent NPF motif peptide, confirming binary complex formation.
**J** Competitive titration experiments show that the non-fluorescent NPF motif peptide can compete with the binding of the labeled one with similar affinity. In vitro binding experiments were carried out in triplicates. **K** Ex vivo measured affinity of HA-POLE2, measured by nHU in cell extract, shows higher affinity than the affinity measured in vitro with purified recombinant protein. **L** The affinity of HA-MBP-POLE2 was found to be weaker than of HA-POLE2 with a clear partial binding activity. Titration nHU experiments were carried out with at least two technical replicates and the obtained standard deviations are displayed for each measurement points to indicate range of measurements. Source data can be found in Supplementary Data 1, 2, Supplementary Figs. 2, 3, 4, 5, and in the Source Data File.

motif, it was rather unexpected to identify multiple types of evolutionarily unrelated NPF motifs as binding partners of the same complex, including the well characterized EH domain-binding NPF motif from SYNJ1. To confirm these findings, the proteome-wide nHU experiments coupled to MS were repeated with the DONSON, WDHD1 and SYNJ1 peptide baits with consistent results and with the additional observation of an interaction between WDHD1 and EHD4 (Supplementary Fig. 2A). The interaction between these distinct NPF motifs and cellular POLE was also validated with targeted western blot analyses (Supplementary Fig. 2B, 3).

### Direct interaction between NPF motifs and POLE2
The experiments revealed that POLE2 consistently showed higher depletion than POLE in binding studies, indicating that the NPF motif

directly interacts with POLE2 and captures POLE indirectly through a reversible POLE-POLE2 interaction (Supplementary Fig. 4). In vitro binding experiments confirmed this direct interaction. Recombinant POLE2, tagged with His$_6$-avi-MBP, was tested with a fluorescein-labeled DONSON_71-88 peptide in fluorescence polarization assays, yielding a dissociation constant of 16 μM (Fig. 1I). Competitive binding assays using a non-fluorescent DONSON_71-88 peptide corroborated this result, with a dissociation constant of 13 μM (Fig. 1J).

In cellular contexts, nHU titration experiments with HA-POLE2 and DONSON_71-88-saturated resin demonstrated an apparent affinity of 0.28 μM, significantly stronger—approximately 40-fold—than the in vitro affinity to MBP-tagged POLE2 (Fig. 1K, Supplementary Fig. 5, Supplementary Data 2). This disparity is likely due to the destabilization of POLE2 when isolated from POLE and the influence of the MBP

tag. Further experiments with HA-MBP-POLE2 in HEK293T cell extracts revealed 10-fold weaker binding compared to HA-POLE2 (Fig. 1L, Supplementary Fig. 5, Supplementary Data 2). HA-MBP-POLE2 also exhibited partial activity where only one out of three molecules could bind the NPF motif.

In summary, despite variations in apparent binding affinity caused by destabilization or the MBP tag, the experiments demonstrated that POLE2 binds directly and reversibly to the NPF motif of DONSON, even in the absence of POLE.

### Structural insight into NPF motif binding by POLE2

We used AlphaFold 3 (AF3) to predict the structure of NPF motif-bound POLE2[36] (Fig. 2). Based on these predictions, the NPF motif of DONSON is bound to POLE2 by adopting a compact Asx turn conformation, similarly to NPF motifs are bound to EH domains or the SSBP2/LDB1 complex. This core NPF motif is bound to a shallow pocket of POLE2, close to its C-terminus. E520 and S522 from the last beta strand of POLE2 are coordinating the amide nitrogen of the Asn sidechain of the NPF motif. The aromatic Phe of the NPF is involved with several hydrophobic contacts, including Y311, that is also in potentially hydrogen bond distance to the carbonyl of the Asn carboxamide side-chain of the NPF motif. Beside the core NPF motif, the N-terminal flanking region also has both main- and side-chain mediated interactions. Y513 from the penultimate beta strand of POLE2 interacts with the amide nitrogen of Asn of the NPF motif. Finally, D284 forms a salt bridge with the NPF motif preceding Arg residue of DONSON.

To validate these predictions, we introduced Y311F, E520A, S522A artificial mutations in HA-POLE2 to disrupt specific interactions of the core NPF motif and expressed these variants in 293 T cells as well. We also introduced D284G, Y513C mutations that are uncharacterized rare, but naturally occurring polymorphic variants of POLE2 located at the putative NPF motif binding interface. Then, prepared cell extracts from each variant and confirmed that none of these mutations caused neither decreased expression of POLE, POLE2, nor degraded products (Supplementary Fig. 6A, B). We performed titration nHU experiments using the DONSON NPF motif as a bait and used dot blot with an antibody against the HA tag to quantify POLE2 depletion (Fig. 2, Supplementary Fig. 7, Supplementary Data 2). This experiment revealed that E520A, S522A, as well as Y513C completely disrupted the interaction and no residual binding could be observed with these variants, confirming that the NPF motif is coordinated by multiple residues from the last and the penultimate beta strands of POLE2. In contrast, the D284G mutation only decreased NPF motif binding by 5-fold and the Y311F mutation did not have any significant impact, showing that Y311 is not contributing to the NPF motif binding via its hydroxy group and that the N-terminal flanking region of the NPF motif can potentially interact with D284 but is not essential for binding. Importantly, both natural POLE2 variants decreased NPF motif binding, classifying D284G as a partial loss-of-function and Y513C as a complete loss-of-function variant for NPF motif binding. Further investigations are needed to evaluate the clinical significance of these natural variants, or similar POLE2 mutations.

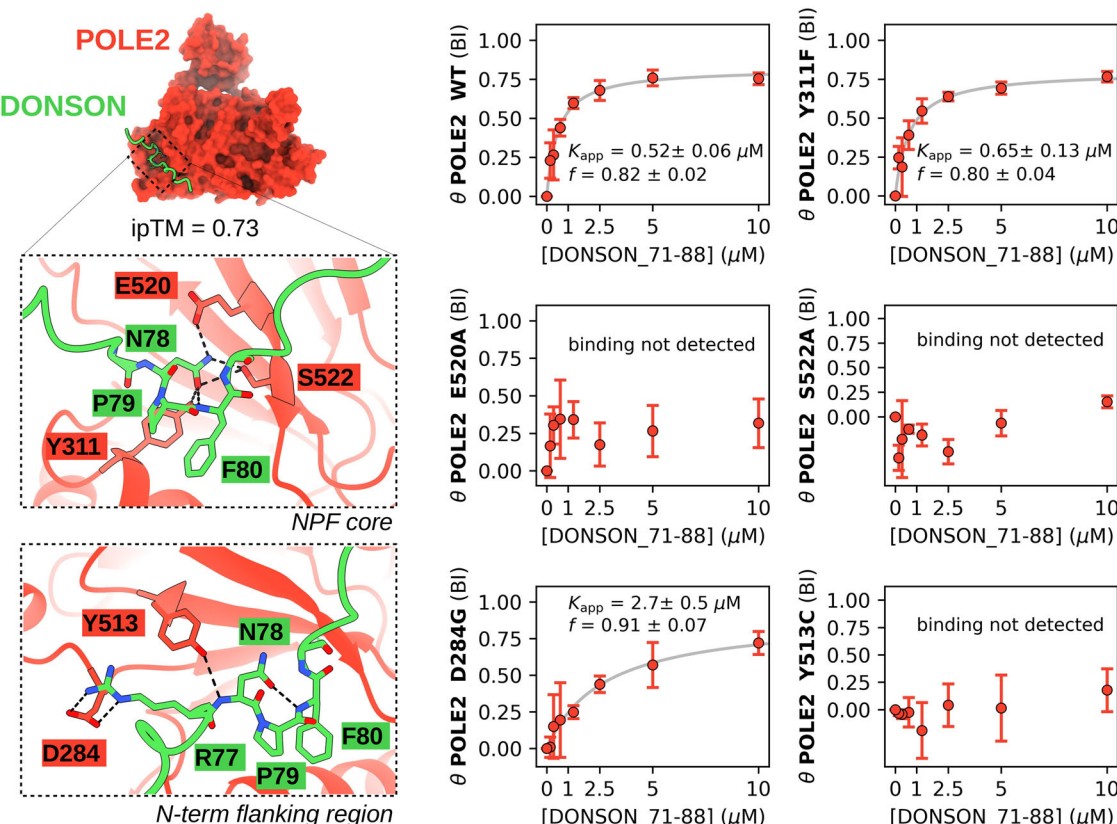

**Fig. 2 | NPF motifs bind to a specific interface on POLE2.** (left) AF3 predictions confidently dock NPF peptides onto a solvent-exposed surface of POLE2 found near its C-terminus. Two insets show directed interactions of the core NPF tripeptide and its N-terminal flanking region. (right) In order to interfere with NPF motif binding, POLE2 residues Y311F, E520, and S522 were selected to be mutated to Ala. In addition, the rare, but natural D284G and Y513C mutations were introduced that are predicted to interact with the N-terminal flanking region of NPF motifs. Titration nHU experiments carried out with these HA-POLE2 point mutants confirmed the predicted bound conformation. Titration nHU experiments were carried out with at least two technical replicates and the obtained standard deviations are displayed for each measurement points to indicate range of measurements. Source data can be found in Supplementary Data 2, Supplementary Figs. 6, 7, and in the Source Data File.

## POLE2 interacts with NPF motifs proteome-wide

So far, our study only focused on a few hand-picked NPF motifs and studied them as short synthetic peptides. However, many other proteins may be present in the proteome that could bind to POLE2 and their putative NPF motifs are embedded in their natural structures that can greatly affect their binding properties. To investigate interactions between POLE2 and intact, full length proteins proteome-wide, we used nHU-MS experiments with recombinant His$_6$-avi-MBP-tagged POLE2 as bait and purified His$_6$-avi-MBP as a control. Even though we are aware that this recombinant POLE2 may bind to NPF motifs with a somewhat altered affinity, we assumed that this would affect all NPF motif-mediated interactions equally, allowing an unbiased affinity ranking of all identified partners. Single point nHU measurements were performed at 26 μM POLE2 concentration and the recovered supernatant was measured by label free proteomic MS measurements (Fig. 3A, Supplementary Figs. 8A, 9, Supplementary Data 1).

In this experiment, we performed an affinity measurement with 6516 endogenous, full length proteins with proteotypic peptides, covering 32% of the human proteome. Out of these detected proteins, 70 showed significant binding to POLE2. Interestingly, no POLE binding was detected during the experiment, possibly due to the competition with endogenous, untagged POLE2 or due to the presence of the N-terminal tag. Nevertheless, the detected full length partners of POLE2 included SYNJ1, NUMB, EPN2, and AGFG1, well known binding partners of EH domains. In addition to these proteins with known NPF motifs, the screen identified WDHD1 as the most significant POLE2 interaction partner, whose isolated NPF motif was studied above. The binding of TTF2 was also identified with high significance, which is a recently reported NPF motif-mediated partner of POLE2. To find other partners containing NPF motifs, we used the SLiMSearch algorithm to find putative NPF motifs in disordered regions of the POLE2 partners and identified 17 proteins with such motifs (Fig. 3B). The NPF consensus sequence is very short, and due to the lack of additional constraints, it is relatively common in the proteome, accounting for 509 NPF motifs in disordered regions of 446 full length proteins out of the approximately 20k human proteins[37]. Based on this, 70 randomly selected proteins are expected to contain only 1 or 2 NPF proteins. In contrast, in our experiments 17 POLE2 partners contained NPF motifs, showing a > 10-fold enrichment in proteins containing NPF motifs in the observed list of interaction partners.

It was previously found that EH domains can bind not only NPF motifs but also to related DPF, GPF, and NPY motifs[38,39]. To investigate the possibility of similar interactions with POLE2, we tested whether the de novo motif discovery tools, such as STREME[40] or SLiMFinder[41] were able to identify recurring motif sequences in the identified POLE2 partners. Unfortunately, these tools failed at this task. STREME mostly identified structural motifs or repeats in the sequences of the identified POLE2 partners, while SLiMFinder[41] did not identify any meaningful motif. Using the sequences of only the disordered regions of these partners the STREME algorithm could identify the NPFQ consensus motif but only as the 15th most enriched motif type (Supplementary Fig. 10A, Supplementary Data 3). Then, we calculated an evolutionary conservation score for each motif instance identified in the STREME output and re-ranked the proposed motifs based on the average evolutionary score of the motifs belonging to the same type (see Materials and Methods) (Fig. 3C, Supplementary Fig. 10B, Supplementary Data 3). This highlighted the NPF consensus motif which was the most conserved among all candidates.

This approach identified 19 motifs in 14 proteins that belonged to this NPFQ motif type. All of them contained the core NPF tripeptide sequence, but they had diverse residues at the last position. No related motif classes, such as DPF or NPY, were identified by this approach. We also carried out binary AF3 predictions, using the full length sequences of POLE2 and its 70 significant interaction partners. This approach identified seven partners that bind to this particular surface of POLE2,

including all four proteins that were well-studied partners of EH proteins, as well as WDHD1 and TTF2 (Fig. 3D, Supplementary Fig. 11). Interestingly, AF3 also predicted interaction with the SEC23-interacting protein (SEC23IP) to the same surface of POLE2 but with an NPY motif. Although this protein is localized to the endoplasmic reticulum and thus spatially separated from POLE2, their intrinsic interaction indicates that the POLE2 binding pocket is compatible with NPY motifs and future studies may also identify similar motifs in other partners.

## Validation of selected nuclear POLE2-NPF motif interactions

Beside the critical role of POLE/POLE2 in replication, POLE is also known to be involved in proofreading, nucleotide excision repair and to a lesser extent in mismatch repair[42]. Similarly, among the identified interaction partners of POLE2, only WDHD1 and TTF2 are known to be involved in replication control, yet many other partners are involved in other nuclear processes, such as transcription regulation or DNA repair. Out of these partners, we selected for further studies WDHD1, TTF2, and ERCC6L. WDHD1 is one of the most significant partners of POLE2 and its NPF motif interacts with POLE/POLE2 with high selectivity. TTF2 – a protein that terminates transcribing RNA polymerase II from the DNA template – contains 2 putative NPF motifs, of which AF3 docked one to POLE2 in binary structure predictions. Two recent preprints already proposed that this NPF motif could play a pivotal role in mitotic replisome disassembly[27,28]. The DNA excision repair protein ERCC-6-like (ERCC6L) DNA helicase contains a single putative NPF motif. The cellular role of ERCC6L is rather understudied and the detected interaction between POLE2 and ERCC6L may be also important for polymerase recruitment during excision repair.

To further investigate these partners, we first synthesized their putative NPF motifs and performed nHU titration experiments using endogenous POLE from cell extracts (Fig. 4A, Supplementary Fig. 12, Supplementary Data 2). As a negative control, we also tested the NPF motif of PYGO2 which showed no binding to POLE (Supplementary Fig. 12). This experiment revealed that the NPF motif of WDHD1 displays a relatively weak interaction with POLE. Out of the two putative NPF motifs of TTF2, only one showed detectable binding with a very strong binding affinity, which also corresponded to the motif that was predicted to bind to POLE according to AF3. No apparent binding can be observed with the NPF motif of ERCC6L.

Then, we cloned these three full length proteins with an HA tag into a mammalian expression vector, produced cell extracts from transfected HEK293T cells, and performed nHU titration experiments using purified His$_6$-avi-MBP-tagged POLE2 as bait and analyzed the results with dot blot (Fig. 4B, Supplementary Figs. 13, 14, 8B, Supplementary Data 2). The affinities measured in this experiment series were in good agreement with those measured by the single-point proteome scale nHU measurements. Indirect binding of these proteins to POLE2 seems unlikely, as only the target proteins were overexpressed in the cells, without any overexpression of potential intermediate partners. Nevertheless, to validate that these interactions are mediated by the identified NPF motifs, we mutated these NPF sites in the sequences of these proteins into KAF motifs which is expected to abolish their interactions with POLE2 and measured the effect of these mutations with nHU experiments. Only the N-terminal NPF motif of TTF2 was mutated that showed much higher affinity as a synthetic peptide motif. Compared to their wild-type counterparts, the binding of all three mutant proteins was drastically reduced. The largest effect could be measured on WDHD1, which showed a 1000-fold weaker affinity to POLE2 upon mutation. The mutated version of ERCC6L also showed a 10-fold weaker affinity to POLE2. The smallest impact was measured in the case of TTF2, which only showed a ~4-fold weaker affinity and could still mediate substantial binding in the presence of the motif-breaking mutation, possibly because of the presence of the secondary weaker NPF motif in its sequence, or because of other

**A**

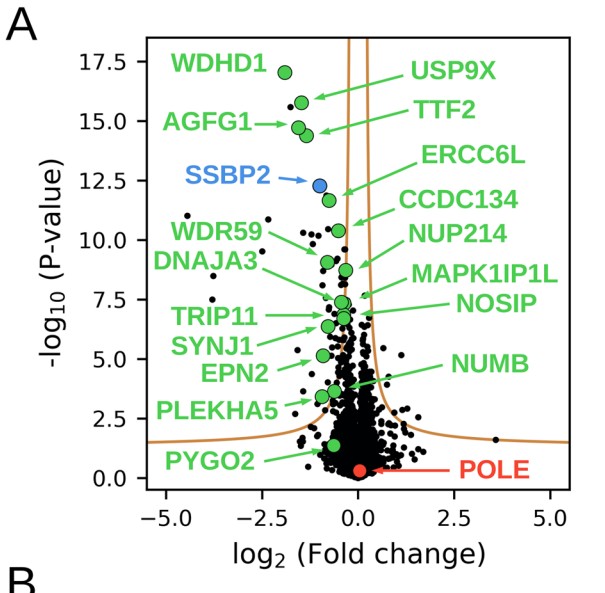

**C**

Partner sequences

IUPred ↓ AlphaFold

Sequences of disordered regions

STREME #15th motif: NPFQ

Re-ranking by average motif conservation

↓

#1st motif:NPFQ

**B**

## Putative NPF motifs in partners

| Protein | θ (BI) | $K_{d.\ app}$ (µM) | putative motif |
|---|---|---|---|
| WDHD1 | 0.73 | 9.4 | GRVNPFKVS |
| AGFG1 | 0.66 | 13 | 4 motifs |
| USP9X | 0.64 | 15 | PLPNPFGDP |
| TTF2 | 0.61 | 17 | 2 motifs |
| SSBP2 | 0.50 | 26 | NFLNPFQSE |
| PLEKHA5 | 0.48 | 28 | LRDNPFRTT |
| EPN2 | 0.47 | 29 | 3 motifs |
| WDR59 | 0.43 | 35 | 2 motifs |
| SYNJ1 | 0.42 | 36 | 3 motifs |
| ERCC6L | 0.41 | 38 | SSINPFNTS |
| NUMB | 0.35 | 48 | SPTNPFSSD |
| CCDC134 | 0.30 | 60 | NFQNPFKID |
| DNAJA3 | 0.26 | 74 | TKHNPFICT |
| NOSIP | 0.23 | 88 | RPLNPFTAK |
| TRIP11 | 0.23 | 88 | TDVNPFLAP |
| MAPK1IP1L | 0.22 | 90 | TPSNPFQVP |
| NUP214 | 0.20 | 103 | ANKNPFSSA |

**D**

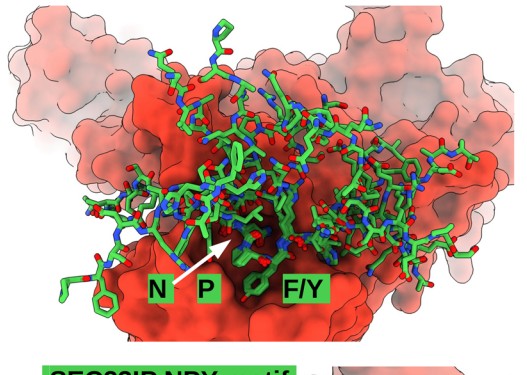

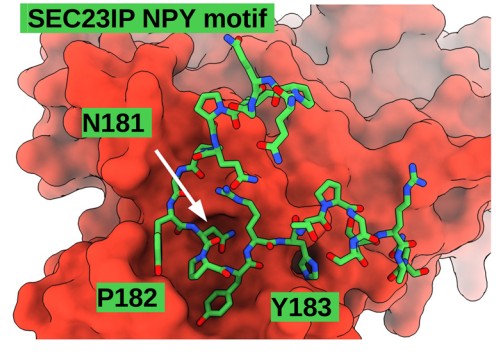

**Fig. 3 | POLE2 binds to NPF motifs proteome-wide. A** Recombinant avi-His₆-MBP-POLE2 was used as a bait in a nHU experiment coupled with mass spectrometry revealing numerous binding partners. Experiments were carried out with 2 biological and 3 technical replicates (total $N = 6$). Two-tailed unpaired T-test, assuming equal variance was used during analysis. Those significantly depleted partners that contain putative NPF motifs, as well as PYGO2 that is not significant but the known partner of the WNT enhanceosome, are highlighted with green or blue circles. **B** List of partners with putative NPF motifs. **C** Schematic pipeline of our de novo motif discovery approach. Based on the sequences of the identified partners, NPF motifs are the most evolutionary conserved ones among all enriched motifs. **D** Superposition of all 7 identified POLE2-bound motifs in AF3 predictions reveal that although most partners bind via a common NPF motifs with a close to identical core motif conformation, the same binding interface is also compatible with a putative NPY motif of SEC23IP. Source data can be found in Supplementary Data 1, Supplementary Figs. 8, 9, 10, 11, and in the Source Data File.

interaction regions. Thus, we could validate the presence of functional NPF motifs in three identified partners of POLE2 which bind to the protein directly.

### Intrinsic connection between distinct NPF motif-mediated networks

Substantial evidence was collected to propose an extensive crosstalk between different NPF motif-recognizing proteins (Fig. 5A). The NPF motif of BORA could not only interact with AURKA, but also with an EH protein, similarly to WDHD1. POLE2 interacted with a putative NPY motif found in SEC23IP, and this protein is a well-known partner of SEC23/24, which is known to be able to bind to NPF motifs[19]. The NPF motif of PYGO2 could not only interacted with the core WNT-enhanceosome – SSBP2/4 and LDB1 – but also with EH proteins and AURKA. POLE2 was also found to interact with the WNT-enhanceosome and with many of their known partners, such as

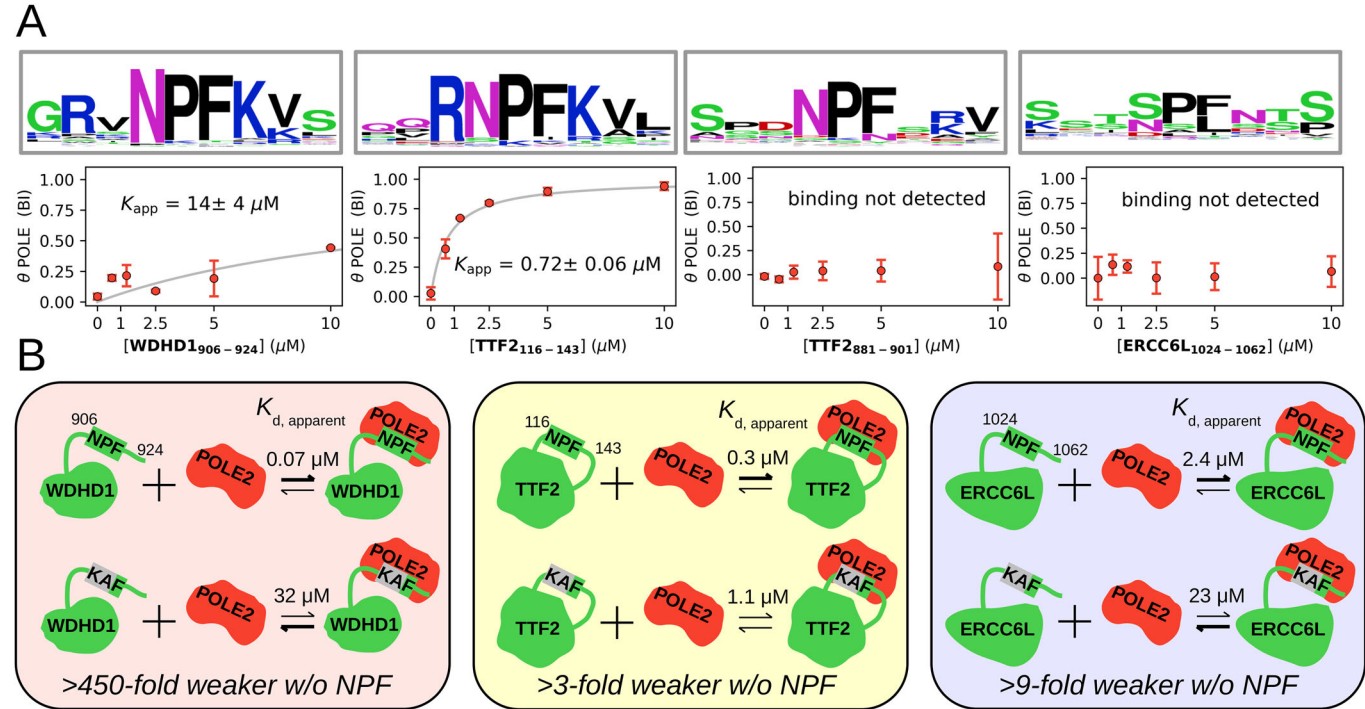

**Fig. 4 | Putative NPF motifs contribute to multiple interactions of POLE2.**
**A** Four NPF motifs taken from three partners were selected for further studies. Below the evolutionary conservation logo of each, the results of titration nHU experiments are shown. Out of these tested NPF motif peptides, only the one taken from WDHD1 and the first motif of TTF2 shows significant binding to endogenous POLE. **B** Full length WDHD1, TTF2 and ERCC6L were expressed in HEK293T cells with HA tags with intact or mutated NPF motifs for nHU experiments. In all three cases, mutations of NPF motifs to KAF sequences significantly decreased their binding strength for recombinant POLE2. Experiments were carried out with at least 2 technical replicates and the obtained standard deviations are displayed on the means for each measurement points to indicate range of measurements. Source data can be found in Supplementary Data 2, 3, Supplementary Figs. 8, 12, 13, 14, and in the Source Data File.

SSBP3, LMO4, LEF1, but not with PYGO2 (Fig. 3A, Supplementary Fig. 9). Yet, the most prominent crosstalk was found between EH proteins and POLE2 as our experiments revealed multiple shared partners between these proteins. Particularly, we found that SYNJ1, a known partner of multiple EH proteins, such as EPS15, could also bind to POLE2. This interaction occurs not only through its isolated NPF motif-rich region, but also as a full length protein. Moreover, other full length binding partners of EH proteins were found to interact with POLE2 in our screen, including NUMB, EPN2 and AGFG1, all of which are NPF motif-mediated binding partners of EPS15[43,44]. Out of these partners, the binding of full length AGFG1 to POLE2 had been identified in a high-throughput yeast-two-hybrid screen[45]. It seems that the interaction networks of these unrelated NPF motif recognition proteins intrinsically tend to overlap.

To investigate the determinants of specificity between POLE2 and EPS15, we further studied the NPF motifs of DONSON (that contains a single NPF motif) and SYNJ1 (that contains three NPF motifs) with nHU titration experiments with total Jurkat extract and monitored the binding of endogenous POLE and EPS15 (Fig. 5B, Supplementary Figs. 15, 16, Supplementary Data 2). We found that POLE binds to the DONSON motif with a micromolar $K_{app}$ and to the triple motif of SYNJ1 - 6-fold weaker, while EPS15 only interacted with the triple motif of SYNJ1 displaying a comparable micromolar apparent dissociation constant. We tried to turn DONSON into an EPS15 binding partner by replacing the three amino acid long flanking regions of its core NPF motif to the flanking region of the central NPF motif of SYNJ1 either on the N-, or C-terminal, or on both sides. We found that replacing either flanking region decreased the apparent POLE binding affinity by ~6-7-fold and when both regions were replaced, the interaction was completely abolished. Thus, POLE2 interaction depends on residues flanking the NPF core sequence. However, none of the DONSON-SYNJ1

chimera motifs gained substantial affinity to EPS15, indicating that either the middle NPF motif of SYNJ1 is not recognized by EPS15, or that efficient EPS15 binding requires additional factors.

We hypothesized that this factor may arise from the multivalent nature of EPS15 interactions. To test this, we created a triple NPF motif peptide using the DONSON sequence, maintaining the same spacing as the NPF motifs found in the SYNJ1 sequence. As expected, the 3xDONSON peptide displayed an increased apparent affinity to POLE/POLE2 due to the increased NPF motif density. Unexpectedly, the 3xDONSON peptide also showed a strong gain of affinity to EPS15, even surpassing the natural SYNJ1 peptide motif ligand in binding strength. Multivalency is almost ubiquitously present within the EH protein family, as EH proteins either contain multiple EH domains or form higher-order oligomeric structures, as observed in the case of EHD proteins[14,15]. Increased apparent affinity of multivalent interaction is a well-known phenomenon and is caused by the increased effective concentration of the interaction sites upon the first binding event[46]. Still, it is rather surprising that the 3xDONSON displayed even a slightly stronger affinity to EPS15 as one of its endogenous partners, SYNJ1, especially given that a single DONSON NPF motif failed to form substantial interaction with EPS15, even when its flanking regions were substituted to one of SYNJ1's. Thus, it seems that even a suboptimal NPF can efficiently bind to EPS15 once its valency is increased to match EPS15's. It is also possible that this NPF motif multivalency contributes to some of those seemingly "off-target" interactions observed between POLE2 and conventional EH domain ligands.

**In vivo proximity interactome of POLE2**
Ex vivo holdup experiments are only capable to uncover the intrinsic properties of interaction networks, as the assay uses purified bait molecules and homogeneous cell extracts. Thus, although we could

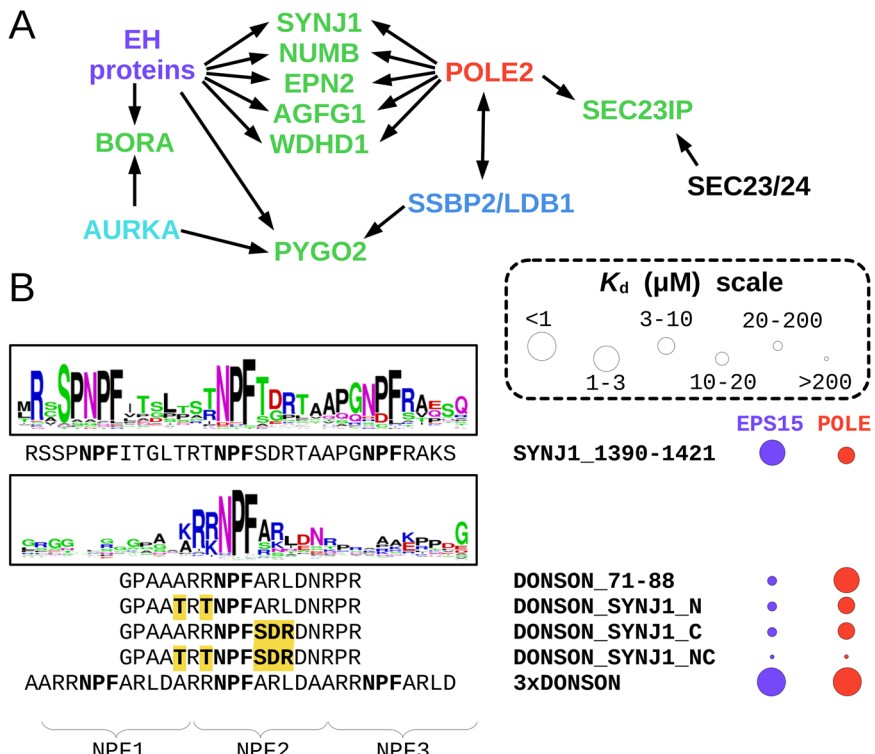

**Fig. 5 | Intrinsic intertwining of discrete NPF networks. A** Many proteins with NPF motifs were found to interact with multiple unrelated NPF receptors and POLE2 was found to interact with many known NPF motif-mediated partners of other known NPF receptors in the presented experiments. This suggests that instead of multiple discrete cellular NPF networks, a single one exists in cells. **B** To understand the determinants of specificity between EPS15 – an EH protein with three EH domains – and POLE2, a series of DONSON chimera motifs were assayed in nHU experiments against binding to both proteins. This experiment revealed the importance of flanking sequences for efficient POLE2 binding and also the importance of multivalency for EPS15 binding. Source data can be found in Supplementary Data 2, Supplementary Figs. 15, 16, and in the Source Data File.

demonstrate that a significant part of the interactomes of various NPF binding proteins can overlap, most of these cross-talks may not occur in their natural cellular environment. To investigate this aspect, we determined the proximity interactome by using proximity labeling mass spectrometry experiments in Flp-In™ T-REx™ 293 cells of WT, as well as mutant POLE2 variants that were used above to validate the NPF motif binding interface (Fig. 6, Supplementary Data 4, Supplementary Fig. 17). Considering all partners of all mutant POLE2, 120 high-confidence interacting proteins were identified that are mostly participating in nuclear processes (e.g. DNA replication and chromatin regulation), but approximately one-third of the partners are linked to cytoplasmic activities (such as vesicle trafficking or cytoskeleton remodelling). This cellular presence outside the nucleus is significant as it strengthens our hypothesis about the possible crosstalk between other cytoplasmic NPF receptors and POLE2. Out of the identified proteins present in the proximity of cellular POLE2, nHU experiments already identified WDHD1, DONSON, PYCR1, GLB1, and EXO1 as POLE2 interaction partners.

By comparing the relative enrichments of partners between WT and mutant POLE2 affecting NPF motif binding, we can clearly identify a set of interaction partners with clear dependence on intact interface. For instance, DONSON only shows binding to WT POLE2 and any mutation at the NPF binding interface eliminates this interaction, even the Y311F. While this mutation does not affect peptide binding in our nHU experiments, it is a structurally sounding disruptive mutation. Surprisingly, EXO1 showed similar effect to the mutations as DONSON, suggesting NPF-dependence. This is unexpected given that the protein does neither contain any NPF motifs nor AF3 is unable to find any reasonable explanation. However, we noted that the interaction between EXO1 and POLE2 was already predicted in the AlphaFold2-

based Predictome project, where an NKF motif, residues 481-483, was found to interact with the same pocket in a similar fashion as NPF motifs[47]. In contrast, WDHD1 binding was found to be mostly unaffected by both the Y311F, as well as the mild D284G mutation, but is significantly disrupted by the other mutations that completely eliminate peptide binding in nHU experiments. Thus, proximity labeling experiment confirmed that some NPF-motif mediated POLE2 interactions are sufficiently stable to be captured by multiple interactomic approaches.

While nHU is a powerful approach to identify transient interactions, it has a limited capacity to capture stable complexes, such as the one formed between POLE and POLE2. In contrary, proximity labeling experiments are more suited to detect such interactions and less sensitive to capture the highly transient ones. Consequently, the proximity labeling experiment revealed strong spatial enrichment of POLE, as well as POLE3 and POLE4 in the proximity of POLE2. In addition to the ε polymerase complex, only a few core replisome proteins were identified as potential POLE2 partners: WDHD1, Claspin, POLA1, DSCC1, and MCMBP. Similarly to WDHD1, both POLA1 and Claspin showed decreased enrichment in the proximity of POLE2 mutants that disrupt NPF binding. POLA1 is a well known interaction partner of WDHD1 and thus this protein may be captured indirectly through WDHD1[48]. However, both Claspin and POLA1 contain a single putative orphan NPF motifs in their disordered protein regions and their direct interaction with POLE2 may be mediated through these sites. In support of this, AF3 successfully docks the putative NPF motif of full length POLA1 onto the correct binding pocket of POLE2, but it does not find a proper solution in the case of Claspin (Fig. 6D). This is in excellent agreement with nHU binding experiments performed with isolated NPF motifs from POLA1 and Claspin that revealed that POLA1

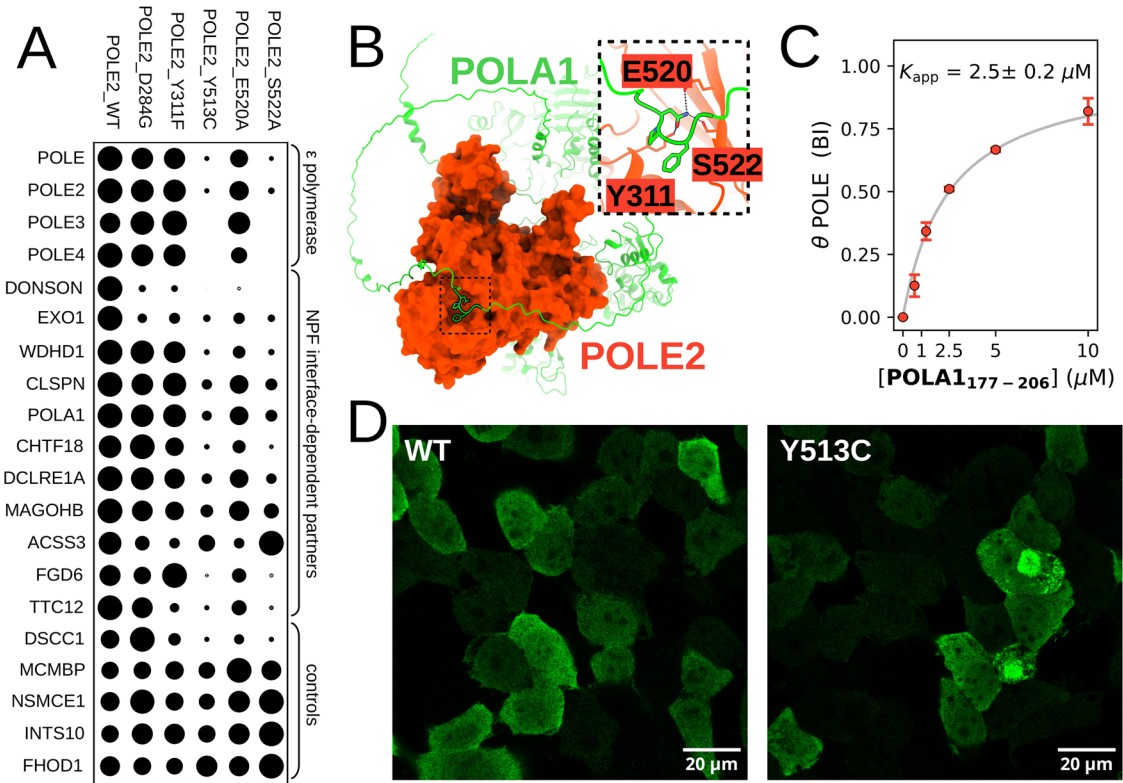

**Fig. 6 | High-confidence in vivo interaction network of POLE2, revealed by proximity labeling mass spectrometry. A** Selected interaction partners of POLE2 are shown in dotplot with the node sizes adjusted based on measured relative abundances. Components of the ε polymerase complex, the partners that are consistently impacted by interface mutations and some reference interactors are shown of which measured abundance does not show change upon mutations. **B** AF3 prediction between POLE2 and POLA1 suggests a potential NPF motif-mediated interaction. **C** nHU experiment confirmed the interaction between the isolated NPF motif of POLA1 and POLE. Experiments were carried out with two technical replicates and the obtained standard deviations are displayed on the means for each measurement points to indicate range of measurements. **D** The Y513C variant that shows decreased stability in proximity labeling experiments localizes to granular cytoplasmic structures that are otherwise absent in cells expressing WT POLE2. Localization experiments were carried out in at least 3 independent experiment series. Source data can be found in Supplementary Data 4 and Supplementary Figs. 17, 18, and in the Source Data File.

could mediate strong interaction with POLE, while the NPF motif of Claspin could not interact with POLE (Fig. 6D, Supplementary Fig. 12, Supplementary Data 2).

Besides revealing important cellular POLE2 interactions mediated by NPF motifs, the proximity labeling experiment series also identified that the disruptive Y513C and S522A mutations reduce POLE2 stability. To further investigate on this, we compared the cellular localization of transiently expressed HA-POLE2 in HEK293T cells (Supplementary Fig. 18). This screen identified that while most mutations do not perturb POLE2 localization, in cells highly expressing POLE2, on rare occasions D284G and much more frequently Y513C localize to large cytoplasmic granules, probably forming aggregates due to the saturation of quality control system[49]. These findings raise important clinical questions concerning those individuals that carry the Y513C mutation resulting in loss of NPF motif binding and reduced POLE2 stability.

## Discussion
### Motif mediated interactions: distinct networks or coherent interactome
Motif-mediated interactions use limited chemical information space. Their core sequences are short, typically spanning just a few residues where only a few amino acid combinations are permitted. This creates interactomic degeneracy where one motif binding protein can interact with large numbers of proteins that all contain similar motifs. Certain types of motifs – usually those that are organized around a stable conformation, such as in the case of NPF motifs – are also recognized

by not only multiple members of the same motif binding protein family, but also by multiple unrelated families of different motif binding proteins. These interaction networks are most often depicted with reductionism as distinct networks with no overlap. However, these networks co-exist in the same cellular environment, and they have an intrinsic driving force to merge because they evolved to capture very similar targets.

In the case of NPF motifs, our study highlights that there is a continuum between the interaction networks of EH proteins, POLE2, and other NPF-recognizing protein. Two NPF motifs were found to interact with multiple types of NPF-motif binding proteins out of the five tested. Additionally, we demonstrated that cross-interactions with the EH protein EPS15 can be easily introduced into the NPF motif of DONSON, which naturally displays high selectivity for POLE2. Moreover, the intrinsic interactome of POLE2 revealed its ability to interact with multiple well-known EH protein partners, such as SEC23/24, and even showed interaction with the NPF-binding protein complex SSBP2/LDB1. Although this intrinsic property to form a coherent network clearly exists, we are yet to find evidence that the interaction actually forms in cells. Spatial separation via compartmentalization can prevent the formation of complexes, even those with high intrinsic affinity and in the case of POLE2, we indeed found that the protein exists in a different spatial environment compared to other NPF motif binding proteins. However, under certain circumstances such interactions can still take place and one should not rule out the possibility of the coherent interactome organized around NPF motifs. In support of this, an interaction was already observed between EPS15 and AURKA, two

NPF motif binding proteins, in HEK293T cells[50] and between POLE2 and AGFG1, a known EH domain binding protein[44,45].

Such interactomic overlap between unrelated motif binding proteins is not unique for NPF motifs. For instance, we previously described that PDZ-binding motifs can be also bound to kinases, phosphatases and 14-3-3 proteins[8], but in principle all motif-mediated interactions that are regulated by post-translational modifications – phosphorylation, acetylation, etc. – display similar interactomic degeneracy. Similarly, many of the known consensus sequences of unrelated motif binding proteins can overlap, that could, in principle, lead to merged interactomes. For instance, Pro-rich motifs that are often adopt PPII type helical conformations can be recognized by a wide variety of domains, including SH3, WW, EVH1, PTB, or EH domains[29]. For this reason, we propose to be cautious when relying on consensus sequences to predict interactions of a given motif sequence as these approaches may lead to ambiguous results.

### NPF motifs involved in mechanisms linked to ε DNA polymerase

The ε polymerase holoenzyme is an essential machinery for our cells, most notably responsible for leading strand synthesis during replication (Fig. 7A). The complex consist of four subunits, the catalytic subunit POLE, the non-catalytic subunit POLE2, as well as two non-essential accessory subunits POLE3 and POLE4 that adopt a histone-like fold. Out of these proteins, most attention was given to the catalytic subunit. This is in part due to the occurrence of multiple clinically observed POLE mutations that lead to Polymerase Proofreading-Associated Polyposis (PPAP), associated with an increased risk of colorectal, stomach, or upper gastrointestinal tract cancers[51]. However, POLE2 also seems to be critical for proper functioning of the holoenzyme. Upon degradation of POLE in U2OS cells, a decreased POLE2 level and S-phase arrest were observed[52]. By introducing the C-terminal non-catalytic domain of POLE in these cells, that directly interacts with POLE2, POLE2 levels are not only increased, but replication initiation was partially restored with delta polymerase being able to partially replace the function of POLE. This indicates that POLE and POLE2 stabilize each other and that the recruitment of POLE2 at the CDC45-MCM-GINS (CMG) helicase is critical for replication initiation, even in the absence of POLE.

In the replisome, POLE2 binds the MCM-GINS junction of the CMG helicase and tethers POLE to the machinery[53] (Fig. 7B). In this conformation, the NPF motif binding surface of POLE2 is solvent-accessible and in the two human replisome structures both POLE2 and WDHD1 are present. Among all POLE2 partners that contain putative NPF motifs, WDHD1 was the one showing the highest affinity and it also showed the highest enrichment in the proximity labeling experiments. WDHD1 is a trimeric hub protein of the replisome and one of its major functions is to tether alpha DNA polymerase for lagging-strand synthesis. Although POLE2 situates on the other end of the replication fork, according to available replisome structures[53,54], the NPF motifs in all three copies of the trimeric WDHD1 disordered tails are easily within reach of POLE2. By looking at the experimental low resolution cryo-electron microscopy density maps of the core human replisome 7PFO or the replisome-CUL2/LRR1 complex 7PLO, unexplained densities can be found on a specific surface of POLE2, but nowhere else in its surrounding region (Fig. 7C). This density perfectly coincides with the core NPF motif that was docked on POLE2 by AF3. Thus, it is very likely that in these experimentally captured replisome states, the NPF motif of WDHD1 is bound to POLE2, possibly posing dynamic constraints between the far ends of the replication fork. Similar constraints could be posed through the interactions mediated with POLA1, but this interaction could also play a role at replication initiation through the primase activity of the DNA polymerase alpha complex. In addition to these partners, DONSON is also a critical replisome protein, essential for CMG helicase assembly. At replication initiation, DONSON connects the pre-loading complex – consisting of TOPB1 and GINS1 – to the DNA-bound MCM-CDC45 complex, forming the CMG helicase. Its conserved NPF motif can be important to loosely tether the ε polymerase holoenzyme to forming replisomes[25,26]. At last, recent studies also identified the NPF motif-dependent interaction between POLE2 and TTF2 and they have found that this interaction is important during mitotic or stalled replisome disassembly by bringing in close proximity a TTF2-bound ubiquitin ligase[27,28].

While many of the identified interaction partners of POLE2 that contains NPF motifs are linked to the replisome, the ε polymerase holoenzyme is also known to be involved in other cellular activities, such as nucleotide and base excision repair due to the exonuclease activity of POLE. In alignment with this, most of the interaction partners identified by our holdup experiment are not directly linked to replisome activities. For instance, we identified ERCC6L, a helicase related to ERCC6, which is known to be involved in nucleotide excision repair. Similarly, the exonuclease EXO1 is known to be involved in DNA mismatch repair. In addition, we also identified multiple binding partners of POLE2 that are related to transcriptional control, such as TTF2 with NPF motifs, or SSBP2, a protein without well-conserved NPF motif, but capable of binding NPF motifs. Such interactions may be important during damage-repair, or for replication-transcription control. While these are certainly lesser-known cellular mechanisms of the ε polymerase, NPF motif-mediated interactions may be just as important as those linked to replication and future work is going to be required to assess their biological significance.

## Methods

### Cloning

Protein coding sequences (POLE2: UniProt ID P56282, residues 1-527; WDHD1: UniProt ID O75717, residues 1-1129; TTF2: UniProt ID Q9UNY4, residues 1-1162; ERCC6L, UniProt ID Q2NKX8, residues 1-1250) were obtained from cDNA pools using standard protocols. For transient transfection, the protein sequences were cloned into the pCI standard mammalian vector containing a HA-tag for N-terminal tagging with standard restriction cloning. The different mutations were introduced by using standard "quick change" site-directed mutagenesis with Platinum SuperFi II polymerase (Thermo Fisher Scientific). Successful mutagenesis was verified by Sanger-sequencing (Eurofins Genomics Germany GmbH, Ebersberg, Germany). For bacterial expression, POLE2 was cloned as $His_6$-AviTag-MBP-TEV-POLE2 construct in a custom bicistronic pET vector that also contains the coding sequence of the *E. coli* BirA enzyme. The empty $His_6$-AviTag-MBP-TEV-STOP vector was used to produce biotinylated MBP for nHU control experiments. For proximity labeling experiments full length POLE2 was cloned to C-terminal (MAC3-tag-C; Addgene Plasmid #185481) destination vector.

### Mammalian cell extract preparation for nHU experiments

Jurkat E6.1 cells (ECACC #88042803, RRID: CVCL_0367) were grown in RPMI (Gibco) medium completed with 10% FBS (Gibco BRL) and 40 µg/ml gentamicin (Gibco/Life Technology). To prepare total cell extracts, Jurkat cells were seeded onto T-175 flasks and grew until $3 \times 10^6$ cells/ml confluency. Cells were collected by 1000 g x 5 min centrifugation, washed with PBS and then lysed in ice-cold lysis buffer (Hepes-KOH pH 7.5 50 mM, NaCl 150 mM, Triton X-100 1%, cOmplete EDTA-free protease inhibitor cocktail 5x, TCEP 5 mM, glycerol 10%).

HEK293T cells (authenticated with 100% match as ATCC ref. CRL-3216, RRID: CVCL_0063) were grown in Dulbecco's modified Eagle's medium (DMEM, Gibco, glucose: 1 g/liter) completed with 10% FCS and 1% Penicillin-Streptomycin (Gibco) on CellBIND flasks (Corning), diluted 1:10 every 3rd/4th day. All cells were kept at 37 °C and 5% $CO_2$. For transient transfection, cells were seeded on poly-D-lysine (Sigma-Aldrich, ref. P6407) treated 100 mm cell culture dishes ($2 \times 10^6$ cells/dish). The next day, the transfection was carried out with JetPRIME reagent (Polyplus) according to the manufacturer's recommendations

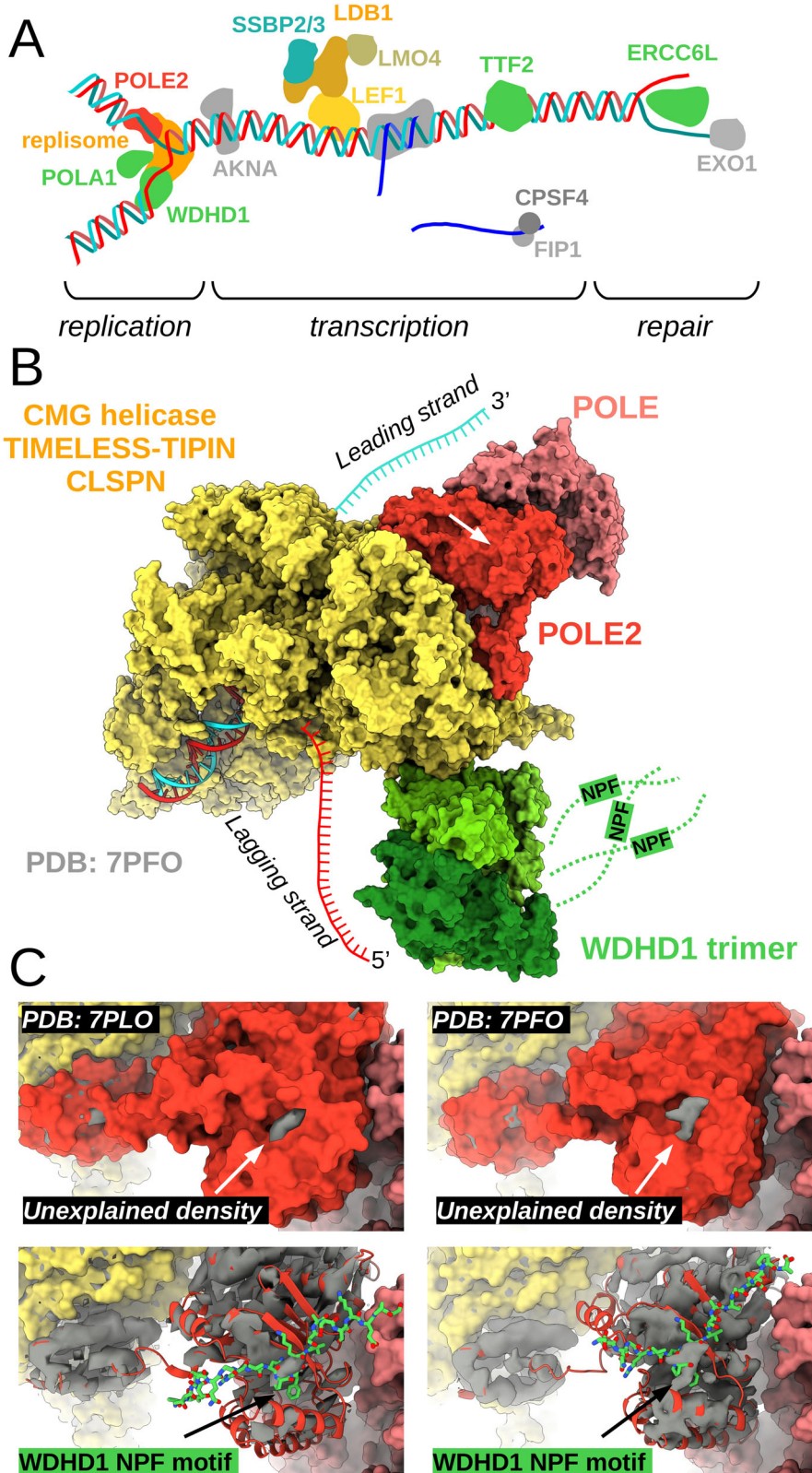

**Fig. 7 | The role of NPF motifs in cellular processes, mediated by POLE2.**
**A** Identified partners of POLE2 are involved in replication, transcriptional regulation, as well as DNA repair mechanisms. **B** In known replisome structures, the relative position of WDHD1 and POLE2 permits the binding of the WDHD1 NPF motifs. Such stapling would pose a flexible, yet constrained tethering of

polymerases responsible for leading and lagging strand synthesis. **C** In the experimental density maps of published POLE2 cryo-EM structures, unexplained densities can be observed on the POLE2 surface perfectly coinciding to the NPF motif sequences docked by AF3.

using 5 µg DNA and 20 µl of JetPRIME reagent/well. The medium was replaced with fresh medium after 5 hours of treatment with the transfection mixture. Two days after transfection, cells were washed with PBS and then lysed in ice-cold lysis buffer (Hepes-KOH pH 7.5 50 mM, NaCl 150 mM, Triton X-100 1%, cOmplete EDTA-free protease inhibitor cocktail 5x, TCEP 5 mM, glycerol 10%).

Cell suspensions were sonicated 4×20 sec with 1 sec long pulse on ice, then incubated rotating at 4 °C for 30 min. The lysates were centrifuged at 12,000 rpm 4 °C for 20 min and supernatant was kept. Total protein concentration was measured by standard Bradford method (Bio-Rad Protein Assay Dye Reagent #5000006) using a BSA calibration curve (MP Biomedicals #160069, diluted in lysis buffer) on a Bio-Rad SmartSpec 3000 spectrophotometer instrument. Lysates were diluted to 2 mg/ml concentration, aliquoted and snap-frozen in liquid nitrogen and stored at −80 °C until measurement.

## Peptide synthesis

The NPF motif peptide of DONSON (residues 71-88, GPAAARRNP-FARLDNRPR) was chemically synthesized on an ABI 443 A synthesizer with standard Fmoc strategy with biotin group attached to the N-terminus via a TTDS (Trioxatridecan-succinamic acid) linker. The NPF motif peptides of SYNJ1 (residues 1390-1421, RSSPNPFITGLTRTNPFSDRTAAPGNPFRAKS), TTF2 (residues 116-143, HHSSNWLRNPFKVLDKNQEPALWKQLIK or 881-901, GRSPNNPFSRVA-LEFGSEEPR), WEE1 (residues 228-247), BORA (residues 18-38, RIPVLNPFESPSDYSNLHEQT), ERCC6L (residues 1024-1062, RRIVSD-GEDEDDSFKDTSSINPFNTSLFQFSSVKQFDAS), PYGO2 (residues 57-86, HLTEFAPPPTPMVDHLVASNPFEDDFGAPK), POLA1 (residues 177-206PVMILKKKRSIGASPNPFSVHTATAVPSGK), CLSPN (residues 1223-1246, SRPMVIQESKSLLRNPFEAIRPGS), and WDHD1 (residues 906-924, SSQGRVNPFKVSASSKEPA), as well as all DONSON chimera peptides were obtained commercially from Synpeptide (SYNJ1 peptide) or GenicBio (all other peptides) with an N-terminal biotin label attached to the N-terminus via an Ahx (6-aminohexanoic acid) linker. The fluo-DONSON peptide (residues 71-88) was N-terminally carboxy-fluorescein labeled without using any linker. All peptides were C-terminally amidated and HPLC purified (>95% purity). Predicted peptide mass was confirmed by MS and peptide concentration was determined based on dry weight.

## Protein expression, purification

POLE2 was co-expressed with BirA in *E. coli* BL21(DE3) cells. At Isopropyl β-D-1-thiogalactopyranoside (IPTG) induction (0.5 mM IPTG at 18 °C for overnight), 50 µM biotin was added to the media. Harvested cells were lysed in a buffer containing 50 mM TRIS pH 7.5, 150-300 mM NaCl, 50 µM biotin, 2 mM 2-mercaptoethanol (BME), cOmplete EDTA-free protease inhibitor cocktail (Roche, Basel, Switzerland), 1% Triton X-100, and trace amount of DNAse, RNAse, and Lysozyme. Lysates were frozen at −20 °C before further purification steps. Lysates were sonicated and centrifuged for clarification. Expressed POLE2 was captured on a HisTrap FF column (Cytiva, 2 × 1 ml), were washed with at least 10 column volume cold wash buffer (50 mM TRIS pH 7.5, 150 mM NaCl, 2 mM BME) before elution with 250 mM imidazole. The Ni-elution was loaded on a pre-equilibrated MBPtrap HP column (Cytiva, 5 ml). The amylose column was washed with 5 column volume cold wash buffer before fractionated elution in a buffer containing 25 mM Hepes pH 7.5, 150 mM NaCl, 1 mM TCEP, 10% glycerol, 5 mM maltose, cOmplete EDTA-free protease inhibitor cocktail. To remove further impurities, POLE2 was ion exchanged with a Q HP column (Cytiva) using a linear gradient from buffer A (50 mM Tris pH7.5, 25 mM NaCl, 2 mM BME) to buffer B (50 mM Tris pH7.5, 1 M NaCl, 2 mM BME) with full length POLE2 eluting in a single peak at ~20% buffer B. Biotinylated MBP was expressed similarly, but was only purified by the two affinity chromatography steps. 10% glycerol, 5 mM TCEP was supplemented to the eluted proteins. The concentration of proteins was determined by their UV absorption at 280 nm before aliquots were snap frozen in liquid nitrogen and were stored at −80 °C.

## Single-point and titration nHU experiments

Single-point nHU experiments with peptide baits were carried out at ~10 µM estimated bait concentration. For this, 50 µl streptavidin resin (Streptavidin Sepharose High Performance, Cytiva) was incubated with 250 µl 70 µM biotinylated peptides (or biotin for controls) in 0.5 ml Protein LoBind tubes (Eppendorf). After mixing, reaction mixtures were incubated overnight at 4 °C, or for 1 h at room temperature. After the incubation, all resins were washed with 9 resin volume (450 µl) holdup buffer (50 mM Tris pH 7.5, 300 mM NaCl, 1 mM TCEP, 22 filtered). The washed resins were then mixed with 25 µl 1 mM biotin solution, diluted in 10 resin volume holdup buffer and were incubated for 5 min at room temperature. Then, the resulting resins were washed four times in total with 9 resin volumes holdup buffer. The resulting bait-saturated resins were mixed with 200 µl 2 mg/ml Jurkat extracts and were incubated at 4 °C for 2 h with constant mild agitation. After the incubation ended, the solid and liquid phases were separated by a brief centrifugation (15 s, 2000 g) and 130 µl of the supernatant was recovered rapidly. Then, to minimize carryover contamination from resin, the recovered supernatants were centrifuged one more time and 100 µl of the supernatant was recovered that was subjected for mass spectrometry analysis.

Single-point nHU experiments with protein baits were carried out nearly identically with two key modifications. First, the resin saturation step was carried out by mixing 50 µl streptavidin resin with 0.5–1 ml 50 µM biotinylated protein solution and the mixture was incubated for 1.5 h at 4 °C with constant agitation. Second, instead of using estimated bait concentrations, we measured it experimentally. An extra aliquot of resin stock was prepared alongside the nHU samples that were identically processed that were used to eluate the immobilized proteins under harsh conditions, in order to experimentally quantify the $C_{bait}$ by using densitometry of Coomassie stained SDS-PAGE that included a protein calibration series. For this, 50 µl bait-saturated resin aliquots were mixed with 200 µl 4x Laemmli Sample Buffer (120 mM Tris-HCl pH 7, 8% SDS, 100 mM DTT, 32% glycerol, 0.004% bromphenol blue, 1% β-mercaptoethanol), instead of cell extract. After 15 min incubation at 95 °C, these samples were analyzed by SDS-PAGE using protein calibration standard, made from recombinant MBP. Densitometry was carried out after Coomassie staining on raw Tif images by using Fiji ImageJ 1.53c and experimental bait concentration were determined by adjusting for molecular weight differences between MBP and MBP-POLE2 constructs. This experiment revealed that the measured bait concentration (26–40 µM) is slightly higher than the empirically estimated (10 µM).

Titration holdup experiments were carried out as described above using 25 µl saturated resins prepared[35]. Briefly, we mixed control, or bait-saturated resins at various proportions by keeping the total resin amount and resin-analyte ratio constant during the experiment (for 25 µl we used 100 µl cell extract as analyte). In titration experiments with protein baits, bait concentration was determined as described above for the saturated resin stocks. Experiments were carried out at 4 °C for 2 h and recovered supernatants were subjected to western blot or dot blot analysis.

## Liquid chromatography–mass spectrometry analysis of nHU experiments

The 100 µl nHU samples were TCA-precipitated by adding 20 µl 100% TCA (#100807, Merck). Immediately after TCA addition the samples were mixed well with vortex and left overnight at 4 °C to precipitate. The following day the samples were centrifuged at 13,000 rpm for 15 min at 4 °C. The protein pellets were washed with 400 µl ice-cold 100% acetone, and centrifuged 13,000 rpm for 8 min at 4 °C. The supernatant was discarded and the pellets washed again with 400 µl

ice-cold 100% acetone followed by centrifugation at 13,000 rpm for 5 min at 4 °C. After centrifugation the supernatant was carefully removed by pipetting and pellets were air dried. To drive off acetone the pellets were placed on a 95 °C heat block for 5-10 min until no liquid was visible. 50 µl 8 M Urea was added to the pellets, mixed with vortexing and put to −20 °C to solubilize. The following day the samples were vortexed and left overnight to 4 °C to solubilize. After the incubation 8 M Urea was diluted with 100 mM Tris-HCl, pH 8.0 to 4 M by adding 50 µl buffer, mixing by vortexing. Proteins were reduced with 5 mM (11 µl of 50 mM stock) TCEP for 30 min at 37 °C, and alkylated with 10 mM iodoacetamide (12 µl of 100 mM stock) for 30 min in the dark at room temperature. Urea concentration was diluted to ~1 M by adding 300 µl 100 mM Tris-HCl, pH 8.0. The pH of the samples was controlled and 1 µg Trypsin/Lys-C Mix (V5073, c = 0.5 µg/µl, Promega) was added to each sample and mixed by vortexing. Samples were digested overnight at 37 °C. Peptide mixtures were then desalted on a C18 mini columns (#HUM S18V, Higgins Analytical, Inc.), and dried in a centrifuge concentrator (Concentrator Plus, Eppendorf).

For MS acquisition the dried peptides were resuspended in 40 µl Buffer A (0.1% (vol/vol) TFA, 1% (vol/vol) acetonitrile (#83640.320, VWR) in HPLC grade water (#10505904, Fisher Scientific). For the DIA analysis, the resuspended peptides were further diluted 1:20 in buffer A1 (1% formic acid in HPLC water). 20 µl was loaded into an EvotipPure (Evosep, Denmark) according to the manufacturer's instructions and run at the same time as three technical replicates.

The samples were analyzed using the Evosep One liquid chromatography system coupled to a hybrid trapped ion mobility quadrupole TOF mass spectrometer (Bruker timsTOF Pro2, Bruker Daltonics)[55] via a CaptiveSpray nano-electrospray ion source (Bruker Daltonics). An 8 cm × 150 µm column with 1.5 µm C18 beads (EV1109, Evosep) was used for peptide separation with the 60 samples per day method (21 min gradient time). Mobile phases A and B were 0.1% formic acid in water and 0.1% formic acid in acetonitrile, respectively. The MS analysis was performed in the positive-ion mode with dia-PASEF method[56] with sample optimized data independent analysis (dia) scan parameters. We performed DDA in PASEF mode from a pooled sample to be able to adjust dia-PASEF parameters optimally. To perform sample-specific dia-PASEF parameter adjustment, the default dia-short-gradient acquisition methods were adjusted based on the sample-specific DDA-PASEF run with the software "tims Control" (Bruker Daltonics). The ion mobility windows were set to best match the ion cloud density from the sample type-specific DDA-runs. The following parameters were used in dia-PASEF runs: TIMS settings 1/K0 0.85–1.30, Ramp time 100 ms, Accumulation time 100 ms, Duty cycle 100%, and dia-PASEF settings cycle time estimate 1.17 s, MS1 ramps 1, MS/MS ramps 10, MS/MS windows 18, mass range 357.2–1257.2 Da, Mobility range 0.85–1.30 1/K0. For collision energy at 0.60 1/K0 20 eV and at 1.60 1/K0 59 eV were used.

To analyze diaPASEF data, the raw data (.d) were processed with DIA-NN v1.8.1[57,58] utilizing spectral library generated from the UniProt human proteome (UP000005640, downloaded 05.04.2022 as a FASTA file, 20358 proteins). During library generation, the following settings were used with fixed modifications: carbamidomethyl (C); variable modifications: acetyl (protein N-term), oxidation (M); enzyme:Trypsin/P; maximum missed cleavages: 1; mass accuracy fixed to 1.5e-05 (MS2) and 1.5e-05 (MS1); Fragment m/z set to 100–1700; peptide length set to 7-30; precursor m/z set to 300–1800; Precursor changes set to 2–4; protein inference not performed. All other settings were left to default.

### Statistical and bioinformatics analysis of the proteomics data for nHU samples

The input file for further DIA data analysis was the DIA-NN Report.pg_matrix. The measured protein intensities were normalized based on median intensities of the entire dataset to correct minor loading differences. For statistical tests and enrichment calculations, not detectable intensity values were treated with an imputation method, where the missing values were replaced by random values similar to the 10% of the lowest intensity values present in the entire dataset. Unpaired two tailed T-test, assuming equal variance, were performed on obtained $\log_2$ intensities. For each experiments, a hyperbolic significance threshold was applied as described before[33,35]. This was calculated as

$$y = y_0 + \frac{-c}{x - x_0} \qquad (2)$$

where $y$ is the $P$ value threshold at fold change of $x$, $c$ is a curvature parameter empirically fixed at 1, $y_0$ is the minimal $P$ value threshold that a significant interaction needs to achieve (defined as 1.3 -$\log_{10}$P), and $x_0$ is the width (2 σ) of the fold-change distribution of all detected proteins, calculated by fitting histograms with a Gaussian curve. Obtained fold-change values were converted to apparent affinities using the hyperbolic binding equation:

$$K_{app} = \frac{(C_{bait} - C_{prey} * \theta_{prey}) * (1 - \theta_{prey})}{\theta_{prey}} \approx \frac{C_{bait} - C_{bait} * \theta_{prey}}{\theta_{prey}} \qquad (3)$$

where $\theta$ denotes the degree of binding, $C_{prey}$ and $C_{bait}$ are the prey and bait molecule concentrations and $K_{app}$ is the apparent dissociation constant.

Note that interactions below this stringent significance threshold may be still biologically important and the thresholds were set to minimize Type I error (false positive). Interactions with weak affinities will result in small depletion values and interaction partners that are difficult to quantify will lead to low statistical significance, both resulting in type 2 error (false negative). Imputation is also a clear source of Type II error. For example, endogenous DONSON did not showed as a significant partner of POLE2. However, it is clear at closer inspection that this is due to the difficulty of DONSON quantification by mass spectrometry in our nHU samples and due to an imputation step of the MS analysis. In this case, only two low abundance peptides of DONSON were detected. Out of these, one peptide was completely absent in the POLE2-depleted nHU samples, and the other displayed an approximately 22% depletion, however this value is based on a single quantification. Nevertheless, this depletion equals to a 92 µM dissociation constant, that is an affinity comparable to the other NPF motif-mediated partners of MBP-POLE2.

### Estimating affinities of indirectly captured complexes

In all nHU-MS experiments with NPF motif peptides where POLE/POLE2 binding was detected, the significance level of POLE depletion was higher. This is possibly due to the larger size of POLE compared to POLE2, causing the easier detection of POLE and more robust quantification by MS. On one occasion, this resulted in POLE2 being measured as significantly depleted, but the depletion fell below our statistical significance threshold. Regardless, in all experiments done with DONSON, WDHD1, or SYNJ1 the degree of binding of POLE2 was found to be consistently higher than of POLE. This information itself strongly implies that the NPF motif interacts directly with POLE2 and that this binding can occur regardless whether POLE2 is bound to POLE, or not. Moreover, this shows that the POLE-POLE2 heterodimer is not an obligate dimer and display some dynamic properties. As POLE shows smaller depletion, it also displays weaker apparent affinity against the same NPF motif baits compared to POLE2. If we assume that POLE binds to POLE2 independently to the POLE2-NPF motif interaction, the measured differences in the degree of binding for POLE and POLE2 can even help us to estimate the apparent binding

strength of the POLE-POLE2 interaction, according to

$$K_{\text{app, POLE}-\text{POLE2}} = \frac{(C_{\text{POLE2}}*\theta_{\text{POLE2}} - C_{\text{POLE}}*\theta_{\text{POLE}})*(\theta_{\text{POLE2}} - \theta_{\text{POLE}})}{\theta_{\text{POLE2}}*\theta_{\text{POLE}}} \quad (4)$$

where $C_{\text{POLE2}}$ and $C_{\text{POLE}}$ are the total POLE2 and POLE concentrations in the extract, and $\theta_{\text{POLE2}}$ and $\theta_{\text{POLE}}$ are the measured depletion values of the two proteins against the same bait. Based on proteomic studies, POLE and POLE2 are expressed in stoichiometric amounts in cells, and they form a heterodimer, approximately at 100 nM cellular concentration[59]. However, our cell extracts can be considered as 100-fold diluted cells resulting an approximately 1 nM concentration for POLE and POLE2. Given this, as well as the MS-based prey depletion measurements, Eq. 1 gives an estimated dissociation constant of the POLE2-POLE interaction between 0.1 and 0.3 nM, calculated for the different NPF motif baits.

As of note, we could identify multiple complexes as binding partners of NPF motifs, or POLE2 by comparing lists of interaction partners with interactomic databases[45,50,59,60] (Supplementary Fig. 9). It would be theoretically possible to estimate affinities for interactions within these complexes in a similar fashion.

## Western and Dot blot experiments

For Western blot, samples were mixed with 4x Laemmli Sample Buffer in 3:1 ratio. Equal amounts of samples were loaded on 6-10% acrylamide-gels (generally 16 µg). Transfer was done into PVDF membranes using a Trans-Blot Turbo Transfer System and Trans-Blot Turbo RTA Transfer Kit (BioRad, #1704275). For Dot blot, nHU samples were directly loaded on Nitrocellulose membranes using 48 Sample vacuum manifold (Cleaver Scientific) following the manufacturer's recommendations.

After 1 h of blocking in 5% milk, membranes were incubated overnight at 4 °C with primary antibodies in 5% milk-TBS-Tween. The following antibodies and dilutions were used: POLE antibody (PA5-78113, Thermo Fisher Scientific, RRID: AB_2736449, dilution: 1:2,000), EPS15 antibody (D3K8R, #12460, Cell Signalling Technologies, RRID: AB_2797926, dilution: 1:1,000), HA antibody (HA.11 epitope tag, Bio-Legend, ref. 901501, RRID: AB_2565006, dilution 1:5,000). Then, the membranes were washed three times with TBS-Tween and incubated at RT for 1 h in secondary antibody (goat anti-rabbit(H + L) #111-035-003 RRID: AB_2313567 or goat anti-mouse(H + L) #115-035-146 RRID: AB_2307392) in 5% milk-TBS-Tween (dilution: 1:10,000). After washing three times with TBS-Tween, membranes were exposed to chemiluminescent HRP substrate (Immobilon, #WBKLS0100) and captured in docking system (Amersham Imager 600 or 800, GE). When membranes were re-exposed to other primary antibody raised in the same species, the membranes were stripped in 30 min with mild stripping buffer [glycine (15 g/liter), SDS (1 g/liter), and 1% Tween 20 (pH 2.2)] to remove primary signal. When membranes were re-exposed to a primary antibody raised in a different species, membranes were treated with 15% $H_2O_2$ for 30 seconds to remove secondary signal. After a 30 min re-blocking in 5% milk-TBS-Tween, membranes were incubated overnight with the following primary antibody and continued the blot as described above. For control, anti-α-tubulin primary antibody (Sigma-Aldrich, ref. T9026, RRID: AB_477593, dilution: 1:5,000) or anti-GAPDH antibody (clone 6C5, MAB374, Sigma-Aldrich, RRID: AB_2107445, 1:5000) were used (in 5% milk-TBS-Tween for 1 h).

Densitometry was carried out on raw Tif images by using Fiji ImageJ 1.53c. Data fitting of titration experiments were performed in QtiPlot and data visualization was done using Matplotlib. For interactions following partial binding, the following formula was used:

$$\theta_{\text{prey}} \approx f*\left(\frac{C_{\text{bait}}}{K_{\text{app}} + C_{\text{bait}}}\right) \quad (5)$$

where the $f$ factor is the "fraction of binding capable prey". (For interactions displaying complete binding, the f value was set to 1, which is identical to the equation shown on Fig. 2A).

## Direct and competitive Fluorescence Polarization experiments

Fluorescence polarization (FP) was measured with a PHERAstar microplate reader by using 485 ± 20 nm and 528 ± 20 nm band-pass filters for excitation and emission, respectively. In direct FP measurements, a dilution series of MBP-POLE2 was prepared in 96 well plates (96 well skirted PCR plate, 4ti-0740, 4titude, Wotton, UK) in a 20 mM Hepes pH 7.5 buffer that contained 150 mM NaCl, 0.5 mM TCEP, 0.01% Tween 20 and 50 nM fluoDONSON peptide. The volume of the dilution series was 40 µl, which was later divided into three technical replicates of 10 µl during transferring to 384 well micro-plates (low binding microplate, 384 well, E18063G5, Greiner Bio-One, Kremsmünster, Austria). In total, the polarization of the probe was measured at eight different protein concentrations (whereas one contained no protein and corresponded to the free peptide). In competitive FP measurements, the same buffer was supplemented with 3.85 µM POLE2 protein to achieve sufficient complex formation to be able to quantify competition event. Then, this mixture was used for preparing a dilution series of the competitor not-fluorescent DONSON peptide and the measurement was carried out identically as in the direct experiment. Analysis of FP experiments were carried out using ProFit, an in-house developed, Python-based fitting program[61]. The dissociation constants of the direct and competitive FP experiments were obtained by fitting the measured direct data with a quadratic binding equation first and by fitting the measured competitive data with a competitive equation, using several obtained parameters from the first fit.

Note that unlike titration holdup experiments, conventional fluorescence polarization experiments are unsuitable for detecting partial binding, because this assay only measures the state of the fluorescent molecules.

## Motif identification and analysis

To find all putative NPF motifs in the proteome, the SLiMSearch 4 program was used using default parameters and the NPF consensus sequence with 5 flanking residues on each end[37]. The obtained list was cross checked with the list of significant POLE2 interaction partners based on UniProt identifiers to identify all partners with putative motifs. To visualize evolutionary conservation of selected NPF motifs, automatic sequence alignments of orthologous sequences were performed using ProViz using metazoan sequences, and the human sequence template and sequence logos were created using WebLogo[62].

The sequences were collected for the POLE2 interactome from UniProt[63], and disordered regions were identified using the following criteria: positions classified as disordered were retained if they did not overlap with predicted Pfam[64] domains and they overlapped with the experimentally verified disordered region based on MobiDB[65] Otherwise, regions predicted as disordered by AlphaFold2-rsa (threshold > 0.582) or when AlphaFold2[66] structures were unavailable, based on IUPred2A[67] scores (threshold > 0.4) were considered as disordered. Regions with less than 5 amino acids were omitted.

Motif discovery was performed on disordered regions using the STREME tool[40] with the following parameters: a minimum motif length of 3, a maximum length of 15, and a maximum of 25 motifs to be identified. Vertebrata conservation data were calculated as described previously and were retrieved from the DisCanVis web server[68]. For each predicted motif, the conservation scores were averaged of their corresponding sites and the predicted motifs were reordered based on the mean conservation score.

## AF3 predictions

Structure predictions of binary complexes were performed using the AF3 online server with default settings[36]. To calculate the complexes of

peptide-bound POLE2 structures, the sequence of full length POLE2 was used together with the NPF motif peptide sequences. To calculate the structures of binary complexes of full length proteins, the sequences of POLE2 was used with each interaction partners, except MYCBP2. All obtained structures were analyzed by evaluating if the NPF motif binding pocket of POLE2 was occupied by the interaction partner with an NPF, or NPF-like motifs, or any other protein region. In all instances when this pocket was utilized for interactions, the pLDDT scores of the docked NPF or NPY motifs were higher compared to their immediate environment (5-10 residue flanking regions) and were similar to the scores of the globular POLE2 regions. The obtained peptide-bound structure of POLE2-WDHD1 was superposed to the C-terminal domain POLE2 of the cryo-EM structures and no further adjustment was made to improve the fit of the superposed NPF motif peptide segment into the experimental density maps.

Interface score (ipTM) analysis showed that identified partners with identified NPF motifs scored among the best, while other partners that bind at different interfaces on POLE2 could also achieve relatively high scores. As of note, ipTM scores of individual motifs docked on POLE2 are higher than of full length proteins, likely due to small binding interfaces and long disordered regions.

As of note, a possible explanation for why AF3 was able to provide a high confidence structure prediction is because a pseudo NPF motif-mediated interaction of POLE2 was already observed as a crystallographic artifact. A crystal structure of POLE2 captured a crystallographic dimer where the C-terminal QGF motif of POLE2 is bound to the NPF binding pocket of the non-crystallographic symmetry-related POLE2 molecule[69] (Supplementary Fig. 19). In this structure, the QGF motif similarly forms a turn conformation, where main-chain of Q525 is jointly coordinated by E520 and S522 from the other molecule and where F527 is bound to the same hydrophobic pocket formed by Y311.

All AF3 predictions are available on the Zenodo platform[70].

## Site-directed mutagenesis of POLE2 variants for proximity labeling experiments

Site-directed mutagenesis of the POLE2 gene was performed to generate five single amino acid substitutions (D284G, Y311F, Y513C, E520A, and S522A). Mutagenesis reactions were carried out using Platinum SuperFi DNA polymerase (Thermo Fisher Scientific) according to the manufacturer's instructions. Mutant constructs were generated by PCR amplification of the wild-type POLE2 entry construct using mutagenic primer pairs designed to introduce the desired nucleotide substitutions.

PCR cycling conditions were as follows: initial denaturation at 98 °C for 30 s; 35 cycles of 98 °C for 10 s, 60 °C for 10 s, and 72 °C for 3 min; and a final extension at 72 °C for 5 min. Following PCR, the parental methylated template plasmid was digested with DpnI for 1 h at 37 °C to eliminate template DNA. The digested products were transformed into *E. coli* Omnimax cells, and colonies were selected on LB agar plates containing ampicillin. Plasmid DNA was purified using the NucleoSpin EasyPure Miniprep Kit (Macherey-Nagel), and all mutations were confirmed by Sanger sequencing.

Verified POLE2 entry clones were transferred into the MAC3-tagged C-destination vector by LR recombination. The resulting expression constructs were sequence-verified and subsequently used for stable cell line generation in Flp-In T-REx cells (Thermo Fisher Scientific) using the pOG44 Flp recombinase system.

The sequences of the mutagenic primers used for each substitution are listed below:

D284G (GAT → GGT) Forward: 5′-GAATAAAGGTGCTATGTTTGTG TTTTTATCTGA-3′ Reverse: 5′-ATAGCACCTTTATTCTCCTCTTCTAGC TGTTT-3′

Y311F (TAT → TTT) Forward: 5′-TGCTGGTTTTTCACCAGCACCTC-CAACCTG-3′ Reverse: 5′-GGTGAAAAACCAGCAAACATTATGCGAAG-3′

Y513C (TAT → TGT) Forward: 5′-AGTTTTTTGTCCTTCTAATAAGA-CAGTAGA-3′ Reverse: 5′-GAAGGACAAAAAACTTTGAATGAAAATCC AC-3′

E520A (GAA → GCA) Forward: 5′-GACAGTAGCAGATAGCAAACT TCAAGGCTT-3′ Reverse: 5′-CTATCTGCTACTGTCTTATTAGAAGGA-TAAAA-3′

S522A (AGC → GCC) Forward: 5′-AGAAGATGCCAAACTTCAAGG CTTTTGC-3′ Reverse: 5′-AGTTTGGCATCTTCTACTGTCTTATTAGAA GG-3′

## Proximity labeling experiments

To generate tetracycline-inducible stable cell line expressing C-terminally MAC3-tagged (with ultraID version of the BioID[71]) POLE2, variants and GFP control were introduced into Flp-In T-REx 293 cells (Life Technologies, Carlsbad, CA)[71]. Approximately $1 \times 10^7$, per replicate, Flp-In T-REx 293 cells stably expressing MAC3C-tagged constructs were induced with 2 µg/ml tetracycline for 24 h to induce the transgene expression, and 50 µM biotin was added to the media 3 h before the harvesting to induce the biotinylation. The cells were washed three times with 1 x PBS and harvested with 1 mM EDTA in 1× PBS. The harvested cells were pelleted using centrifugation, snap frozen in liquid nitrogen, and stored at −80 °C. The samples were then suspended in 1.5 ml of lysis buffer (0.5% IGEPAL, 50 mM Hepes, pH 8.0, 150 mM NaCl, 50 mM NaF, 1.5 mM NaVO₃, 5 mM EDTA, 0.1% SDS, 0.5 mM PMSF, protease inhibitors, and 80 U/ml benzonase nuclease (Santa Cruz Biotechnology)) on ice. Lysis was followed by incubation on ice for 15 min and three cycles of sonication (3 min) and incubation (5 min) on ice. The samples were cleared by centrifugation, and the supernatants were poured into microspin columns (Bio-Rad) that were preloaded with 250 µl of (50 % slurry) Strep-Tactin beads (IBA GmbH) and allowed to drain under gravity. The beads were washed 3 times with 1 ml lysis buffer and then 4 times with 1 ml HENN (50 mM Hepes pH 8.0, 5 mM EDTA, 150 mM NaCl, 50 mM NaF) buffer. The purified proteins were eluted from the beads with 700 µl of wash buffer containing 0.4 mM biotin. To reduce and alkylate the cysteine bonds, the proteins were treated with a final concentration of 5 mM TCEP (tris(2-carboxyethyl) phosphine) 20 min at 37 °C by shaking and 10 mM IAA (iodoacetamide) for 20 min in dark. Finally, the proteins were digested into tryptic peptides by incubation with 1 µg sequencing grade trypsin (Promega, V5113) overnight at 37 °C by shaking. The digested peptides were purified using C-18 microspin columns (Higgins Analytical, Inc.) as instructed by the manufacturer. For the mass spectrometry analysis, the vacuum-dried samples were dissolved in buffer A (1% acetonitrile and 0.1% trifluoroacetic acid in MS-grade water).

## Liquid chromatography–mass spectrometry analysis of proximity labeling experiments

The Evosep One liquid chromatography system was utilized to analyze desalted peptide samples. It was coupled to a hybrid trapped ion mobility quadrupole TOF mass spectrometer (Bruker timsTOF Pro 2) via a CaptiveSpray nano-electrospray ion source. Peptides were separated using an 8 cm × 150 µm column packed with 1.5 µm C18 beads (EV1109, Evosep) utilizing the 60-SPD (sample-per-day) method with a 21 min gradient time. Mobile phases A and B were prepared as 0.1% formic acid in water and 0.1% formic acid in acetonitrile, respectively. For all proximity labeling samples, mass spectrometry (MS) analysis was performed in positive-ion mode using a data-dependent acquisition (DDA) strategy in PASEF (Parallel Accumulation Serial Fragmentation) mode, employing the DDA-PASEF-short_gradient_0.5s-cycletime method.

The raw data files were processed utilizing FragPipe v23.1 in DDA+ mode with MSFragger-4.3[72] using human protein fasta files containing 40,970 entries (20,485 decoys: 50%) from the UniProtKB database (downloaded on September 8, 2025). Carbamidomethylation of cysteine residues was used as static modification while amino-terminal

acetylation and oxidation of methionine were used as the dynamic modification. Biotinylation of lysine and N-termini were set as variable modifications. Trypsin was selected as the enzyme with allowance for up to two missed cleavages. The instrument and label-free quantification parameters remained at their default settings. Lastly, the results output consisted of PSM values derived from peptides with a false discovery rate (FDR) below 0.01 as generated by Philosopher.

## Identification of high-confidence protein-protein interactions of proximity labeling experiments

In-house python script, incorporated with Significance Analysis of INTeractome (SAINT) -express version 3.6.3[73] was used as a statistical approach for identification of specific high-confidence interactions from proximity labeling. High-confidence interactions (HCIs) were defined by an estimated protein-level Bayesian FDR (BFDR) of 0. Interactions that passed the filter with any bait were rescued for other baits as well. Furthermore, we used our own in-house contaminant GFP library and CRAPome database (version 2.0)[74]. Preys detected in over 20% of the GFP library or CRAPome samples were deemed contaminants and removed unless their spectral counts in the sample runs were over 3 times higher than the library average. Preys with an average spectral count of less than three were also removed.

## Reporting summary

Further information on research design is available in the Nature Portfolio Reporting Summary linked to this article.

## Data availability

The raw LC-MS/MS data have been deposited to the ProteomeXchange via the MassIVE database with the identifier PXD061783. The AF3 predictions are available at Zenodo (https://doi.org/10.5281/zenodo.16792938)[70]. Source data are provided with this paper.

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

## Acknowledgements

We thank the staff of the IGBMC Cell Culture Facility for their help in cell culturing. G.G. was supported by Inserm, by the ATIP-Avenir program for young group leaders and by the T-ERC STG 2024 (ANR-24-ERCS-0002). This work is part of a project Fondation ARC, funded by Fondation ARC pour la recherche sur le cancer (category PJA1, to G.G., 2024). The project was supported by the EKÖP-KDP-24 University Excellence Scholarship Program, the Cooperative Doctoral Program of the Ministry for Culture and Innovation, and the National Research, Development and Innovation Fund (to NoD and Z.D.). This work has been supported by the French government, through the France 2030 investment plan managed by the Agence Nationale de la Recherche, as part of the Université Côte d'Azur's Initiative of Excellence, reference ANR-15-IDEX-01.

## Author contributions

S.K., D.M.P., K.S., A.T., TT and MV designed and performed proximity labeling and all mass-spectrometry measurements and their data analysis. BZ and GG performed all nHU and in vitro experiments and analyzed data. NDe and ZD performed bioinformatic analysis. NDa helped in experimental design. MV and GG conceived the project. GG wrote the original draft, and all authors reviewed the manuscript.

## Competing interests

The authors declare no competing interests.
