## [Transparent Peer Review file · Nature Communications]

The non-catalytic DNA polymerase ϵ subunit is an NPF motif recognition protein

Corresponding Author: Dr Gergo Gogl

Version 0:

Reviewer comments:

Reviewer #1

(Remarks to the Author)

The paper "The non-catalytic DNA polymerase subunit is an NPF motif recognition protein" focuses on systematic identification of proteins acting as receptors for the three amino acid long NPF-motif. Keskitalo and Zambo et al. utilized a combination of biochemical assays, a set of mutants and structural predictions to screen for NPF-motif recognition proteins. For this well characterized and understudied/orphan NPF-motif containing peptides were used employing a quantitative MS-based hold up assay. They identified POLE2 as a reoccurring binding partner of multiple NPF-motifs. Affinity screens confirmed interactions of POLE2 with nuclear proteins containing NPF-motifs. PPI-database driven network analysis and a proximity interaction mapping approach were used to confirm the role POLE2 as a hub in the NPF-motif containing protein network.

SLiMs are difficult to study, as they are often in IDRs, typically only a few residues long, and lack structural folds. Therefore, most SLiMs currently listed in databases have been predicted using consensus sequence-based rules. To improve SLiM prediction algorithms, it is crucial to expand experimental data on SLiMs to reveal their role in protein-protein interactions. The Authors applied a combination of affinity-based assays (native Hold Up assays), sequence-based (motif) conservation predictions, mutational screens and structural predictions to identify NPF-motif receptors. This generates a useful resource that should be integrated into prediction algorithms of SLiMs, thus enhancing their performance.

The rationale of the experiments, the applied methods and the obtained results are mostly detailed and understandably described (see comments for a few suggestions). Overall, the work represents a significant effort to produce robust and comprehensive analyses of NPF-motif receptors. The figures describing this are mostly clear and convey useful and interesting insights. The authors have also been very up-front about the limitations of their employed methods/approaches (e.g.: Ex vivo holdup experiments are only capable to uncover the intrinsic properties of interaction networks, ...), which is appreciated and should be helpful to readers to judge the reported findings.

I congratulate the authors for their rigorous work on mapping novel NPF-motif receptors. I have only a few minor comments where I think a modest amount of rephrasing, clarification and data analysis could be helpful and I have no trouble recommending this manuscript for publication.

1. It is very much appreciated that raw/minimal processed western blot images (e.g.: Supplementary figures) are provided a long side the WBs and quantitative data shown in the main figures. Further, I want to acknowledge that the MS-raw data was deposited. However, the MS-data was deposited in a difficult to access way (controls and samples packed in *.zib folders with 200 – 300 GB). MassIVE supports the submission of individual raw files and promotes this data structure. Transferring a single packed raw file folder via FTP would have taken me several days, which is unacceptable and makes reusing the files tedious. Thus, I was not able to check the DIA-NN files completeness (raw files, parameter files, sample file list annotation, peptide or protein level outputs) nor the BioID files (raw files, MSFragger (FragPipe) parameter files, outputs). To facilitate easier access, please make the files available individually and provide sample annotation files.

2. I would encourage the authors to make the best performing models of the AlphaFold predicted structures (*.cif or *.pdb files) including the predictions of full-length complexes, accessible to the public by depositing the structures (e.g. Zenodo or a similar platform), including again annotation files.

3. Introduction: "Two of these EH proteins contain three EH domains, four of them contains two domains and the four EHD proteins that contains a single EH domain form stable oligomers 14,15." "contain" instead of "contains". "Since all members of the EH family recognize the same structural motif, a putative NPF motif can interact with any members of the family." "any member" instead of "any members".

4. "Identification of POLE/POLE2 as a recurring interaction partner of unrelated NPF motif containing peptides"; Section: "... unexpected to identify multiple types of evolutionarily unrelated NPF motifs as binding partners of same complex, including the well characterized..." "the same complex" or "same complexes" instead of "same complex".

5. "Structural insight into NPF motif binding by POLE2"; Section: To maintain consistency, please standardize the abbreviation throughout the document. Use either "AlphaFold 3 (AF3)" or "AlphaFold" consistently in all sections.

6. "POLE2 interacts with NPF motifs proteome wide"; Section: "In this experiment, we performed an affinity measurement with 6516 endogenous, full-length proteins, covering 32 % of the human proteome." There are 7,236 Protein Groups reported in Table S1, and 6,516 are matched with proteotypic peptides. Also, the Protein Groups which have multiple mapped UniProtKB identifiers are endogenous, full-length proteins. Rephrase and clarify that only proteins for which proteotypic peptides were found are included in the count of 6516 proteins.

7. "Validation of selected nuclear POLE2-NPF motif interactions"; Section: "Then, we cloned these three full-length proteins with an HA tag into a mammalian expression vector, produced cell extracts from transfected HEK293T cells and performed nHU titration experiments using purified His6-avi-MBP-tagged POLE2 as bait and analyzed the results with dot blot (Figure 4B, S13, S14, S8B, Table S2)." The result section discusses three selected proteins (TTF2, ERCC6L and WDHD1), but Supplementary Figure S14 references two additional proteins, PYGO2 and CLSPN which were not discussed in this section of the manuscript. Is this link wrongly set or was the text adapted to three candidates which worked? This is confusing, especially as PYGO2 and CLSPN were not significant in the POLE2 nHU-MS experiment shown in Fig. 3A (see also comments for Figure 3A).

8. "In vivo proximity interactome of POLE2": Section: "By analyzing the cellular localization of the identified partners, we found that all of them are located either in the nucleoplasm, or in the centrosome, indicating a nuclear and a cytoplasmic POLE2 cellular pool (Figure 6B)." Was GSEA performed (GO CC) or were the localizations manually curated (in Figure 6B, the label for the legend is missing). What thresholds for the GSEA were considered to draw the conclusion regarding localizations? Please clarify.

9. "In vivo proximity interactome of POLE2": Section: "We also find a particularly strong enrichment of proteins involved in biological processes related to DNA replication, DNA damage response, and cell cycle control, processes in which POLE has known functions (Figure 6C)." What GSEA test was used? What gene sets/ terms were used (GO BP?). What thresholds were used? What does "strong" enrichment mean?

10. General comment regarding GSEA/ GO enrichment: GSEA is not covered in the methods. Please consider adding a brief paragraph to the method section.

11. Discussion: "Out of these, POLA1 is an interaction partner of WDHD1 and may be captured indirectly." Lysis conditions for BioID experiments (0.1% SDS) were not native. BioID captures direct or indirect proximity interaction partners, as the enrichment of interactors depends only on the distance to the tagged bait. In BioID direct or indirect can only be inferred using database or literature knowledge.

12. Methods: "Single-point and titration nHU experiments":
Was Benzonase/DNAse treatment performed on Yurkat cell extracts? Benzonase treatment was performed for the BioID experiment in Hek293 cells. If no Benzonase was added, can you exclude co-depletion of proteins through DNA stretches? In the manuscript you report that POLE2 was most likely depleted through its interaction with POLE. How can you rule out that other nuclear/DNA binding proteins were not only depleted through their binding to a DNA-strand on which a NPF-motive receptor protein is too binding?

13. Methods: "Statistical and bioinformatics analysis of the proteomics data for nHU samples":
"The input file to further dia data analysis was the DIA-NN Report.pg_matrix." Rephrase ("to" -> "for")?

14. Methods: "AlphaFold predictions": "In all instances when this pocket was utilized for interactions, the pLDDT scores of the docked NPF or NPY motifs were outstanding compared to their environment and were similar to the scores of the globular POLE2 regions." Either rephrase or provide a comparison (e.g. pLDDT scores of the NPF and NPY motif versus the rest of the protein/peptide sequence). Outstanding cannot be judged with the information provided in the manuscript/and or methods. Another option would be to mutate the NPF or NPY motifs (e.g. to KAF), perform AF3 predictions and check for significant ("outstanding") difference in the scores vs. the peptides with NPF or NPY motifs. Further, the pLDDT is a measure for the confidence in the local structure for each residue. Please also provide the ipTM scores, which is a measure for the relative position of the two chains. A table with all AlphaFold models and scores would be useful to judge the significance of the finding.

15. Methods: BioID experiments: "... and 50 μ M biotin added to 3 h before the harvesting to induce the biotinylation .." added to the media missing?

16. Methods: "Identification of High-Confidence Protein-Protein Interactions of BioID experiments": "Interactions that passed the filter with any bait were rescued for other baits as well." Were any additional baits used for BioID besides POLE2? If not, please rephrase or remove the statement. If there were additional baits use, they were not reported in Table S4.

17. Methods: "Identification of High-Confidence Protein-Protein Interactions of BioID experiments": "High-confidence interactions (HCIs) were defined by an estimated protein-level Bayesian FDR (BFDR) of ≤ 0.01 ." Table S4 reports 48 are proximity interactions. In Fig. 6A only 24 are shown (see also Fig.6 comments). It seems like there are additional 22 PPIs with a BFDR larger 0.01 and smaller or equal to 0.05 reported. Maybe the wrong version of the table was uploaded? Either remove the additional proximity interactors or discuss them in the manuscript and adapt the filtering threshold. To ensure full transparency it would be appropriate to report the full unfiltered scoring output containing background preys and scored proximity interactors.

18. Figure and Figure legends: Figure 1A: "multiple types of receptors" – the proteins shown are according to your manuscript and UniProt not classical receptors. Please ensure that "NPF-motif receptors" is specified in all instances where "receptor" is mentioned in the manuscript, as done in the figure caption of Fig. 1C-H.

19. Figure and Figure legends: Fig.1C-H. I struggle to comprehend why non-significant proteins are highlighted in the same manner as significant NPF-motif recognizing proteins. For example, in Fig.1C, ITSN1 (not significant) is highlighted and labeled, close to it is another highlighted point but not labelled. I suggest using a different label/color, or ring not filled out for non-significant but still discussed examples, instead of a full point. State statistical thresholds (\log_2FC and p-value) in the figure legend and/or main manuscript (section "Identification of POLE/POLE2 as a reoccurring interaction partner of unrelated NPF motif containing proteins"). Highlighted points which have close to no change, for example for WEE1_228-447 (Fig. 1F), BORA_18-38 (Fig.1G) should be differentiated to significant hits.

20. Figure and Figure legends: Figure 2: "AlphaFold predictions confidently dock NPF peptides onto a solvent-exposed surface of" Provide the ipTM-score to support your statement (see also method statement for structure predictions with AlphaFold).

21. Figure and Figure legends: Figure 3: A) "Those significantly depleted partners that contain putative NPF motifs are highlighted with green, or blue circles." Why is PYGO2 highlighted? It does not seem to be significant (\log_2FC -0.63 and $-\log_{10}(p\text{-value})$ of 1.36). Was it highlighted because it contains the NPF motif? If so, rephrase the figure caption for clarity. Further, you state in the section "Intrinsic connection between distinct NPF motif-mediated networks" that there is no interaction. This needs to be revised.

22. Figure and Figure legends: Figure 3: The upper panel is challenging to read. How many putative motifs of partners were overlaid? Please provide the number (N). Are these the 7 motifs mentioned in the main manuscript text and shown individually in Supplementary Figure S.11? I advise showing only one individual structure. Consider pushing the overlay to the supplementary figure instead. Please provide AF scores (e.g. pIDTT, ipTM, pTM) in a supplementary table or add them to the figure caption.

23. Figure and Figure legends: Figure 6A: "A) Identified partners of POLE2, ranked by their enrichment." The plot shows average signal (Average Spectral Counts) and not enrichment against GFP controls. Rephrase statement or change labels. In the main manuscript 26 proximity interactors are mentioned (...the BioID experiment revealed 26 significantly enriched partners ...). In the panel only 25 are shown, and comparison with Table S4 shows that TUBB4A was removed. Why? Add the number of replicates and the number of GFP-controls used to the figure legend.

24. Figure and Figure legends: Figure 6C: To improve clarity, please add a color key for the different node coloring used to the legend. Additionally, the varying edge thickness should be explained in the legend of the plot, indicating what it represents. Also, specify if a significant threshold was used to filter GO terms in the figure legend.

25. Supplementary Figures: Fig. S1B: provide reference to database form where the domain architecture was obtained (InterPro, UniProtKB).

26. Supplementary Figures: Fig. S2B: What is the number of replicates (n=2 biological replicates for each of the 2 blots shown in Figure S3?) for the bar plots shown in panel B? Does the plot show mean, and standard deviation? Clarify.

27. Supplementary Figures: Fig. S9: Explain the color code that the figure is readable (same color code as in Figure 3?).

Reviewer #2

(Remarks to the Author)

This manuscript by Keskitalo et al investigates the role of the NPF SLiM motif in mediating protein protein interactions. They use an approach that they developed called native holdup to screen for binding partners of several different NPF protein including two factors involved in DNA replication (DONSON and WDHD1). One of the top hits from the screens is a subunit of DNA polymerase epsilon called POLE2. A second screen is performed using POLE2 as bait that identifies a number of NPF containing proteins. Mutational analysis based on alphafold 3 predictions convincingly locates the NPF binding site on POLE2. BioID with POLE2 is then used as an alternative method to assay putative POLE2 binding partners. However, this approach only identifies 3 proteins that were also found by nHU. It is then shown that POLA1, which is seen only in BioID, binds to POLE2 also via an NPF motif interaction that is predicted by alphafold.

This is a nice study that helps to define POLE2 as an NPF binding protein. The use of an unbiased approach to identify binding partners of POLE2 demonstrates that POLE2 can bind to diverse NPF containing proteins, many of which are not thought to be involved in DNA replication. However, most of these proteins were not detected by BioID, casting doubt on whether their binding to POLE2 has any biological relevance. The manuscript clearly shows that DONSON, TTF2, WDHD1 and POLA1 can all bind to POLE2 via an NPF-mediated interaction. This has already been shown in published work for DONSON and in recent pre-prints for TTF2. That being said, this manuscript nicely illustrates that there is likely to be competition at replication forks for a single NPF binding site on POLE2. Overall, the experiments are well performed and the data are appropriately presented. Due to a lack of any new functional data the findings presented are a little limited because it remains unclear if the binding to POLE2 to diverse NPFs (beyond known DNA replication factors) has any function. New insights in this direction would enhance the impact of the work. Some specific comments to consider are listed below:

Specific comments

Figure 1 – it might be nice to see peptide sequences. What was the rationale for the sequence choice? e.g. total length, number of residues either side of the NPF?

The authors claim that it is surprising that the NPF motifs of DONSON and WDHD1 bound to POLE2. Given that the Walter lab showed that DONSON binds to POLE2 via this motif in a manuscript published in 2023, and the WDHD1 POLE2 interaction is one of the top hits in the predictomes.org database for WDHD1, the result does not seem particularly surprising.

The measurements of the DONSON NPF binding to POLE2 are quite variable across different experimental set ups, as the authors acknowledge. Therefore, because binding of this region of DONSON to POLE2 has previously been demonstrated, the experiments don't add much new insight.

The AlphaFold prediction of the DONSON POLE2 interaction is described in some detail, perhaps too much detail given that it is based on an in silico prediction and not an experimentally derived structure.

The results of experiments with POLE2 variants are interesting. It is concluded that the Y513C variant abolishes NPF binding. How frequently does this variant occur and is it associated with any phenotypes? Generally, a bit more background here would be helpful. Loss of binding of POLE2 to NPF factors could interfere with multiple important replication processes that could each have phenotypic consequences.

When TTF2 is introduced, the description should be more precise and the relevant pre-prints cited. I think the work from the Labib and Walter / Pellman labs clearly shows that TTF2 is needed for mitotic replisome disassembly and that this requires the NPF-mediated interaction of TTF2 with POLE2. The description in the discussion is more appropriate and could be moved forward.

Are any of the EH binding proteins that were found to bind POLE2 seen in any of the iPOND data sets? This analysis would be helpful because it might provide some insight into whether any of these proteins are ever found at or near to active replication forks.

The authors say that Exo1 was identified in nHU and BioID as a POLE2 binding partner. However, Exo1 does not appear in the list of proteins shown in Figure 3 and there doesn't seem to be any further discussion about Exo1.

The authors assume that the interaction of POLE2 and WDHD1 is "of critical cellular importance". However, they provide no data pertaining to the function of this interaction in any aspect of DNA replication and therefore it still needs to be established if the interaction is critically important.

This work establishes nicely that at least 5 replication proteins (WDHD1, EXO1, TTF2, POLA1, DONSON) bind to the same site on POLE2. This suggests a competition between these factors for binding which is an interesting point for discussion. For example, how does TTF2 bind for replisome disassembly when presumably WDHD1 and POLA1 are still present at forks? Is binding regulated, perhaps via PTMs?

Reviewer #3

(Remarks to the Author)

In this study, Keskitalo et al demonstrate that POLE2, the non-catalytic subunit of human DNA polymerase epsilon, acts as a receptor for NPF-motif-containing peptides. The authors identify multiple NPF-containing peptides and nuclear proteins that interact with POLE2, validating some interactions through biochemical assays and BioID experiments. Based on these findings, they propose that POLE2 serves as a central hub connecting DNA replication with other nuclear processes via its NPF-binding pocket. While the study presents novel and intriguing observations regarding POLE2's role in nucleating NPF-containing factors, the evidence supporting a direct functional link between POLE2-mediated NPF interactions and DNA replication remains incomplete. The mechanistic and functional insights are currently limited. With additional experimental validation, the manuscript could be strengthened. In its present form, it may be better suited for a specialized journal focusing on protein-protein interactions or DNA replication machinery.

Major Concerns:

1. While the interactions are well-documented, the functional relevance of POLE2's NPF-binding activity remains unclear. Additional experiments would help establish whether these interactions are biologically meaningful.
2. To better assess the importance of NPF binding, the authors should generate POLE2 mutants defective in NPF-motif recognition and analyze their impact using mass spectrometry to identify lost interactions.
3. The claim that POLE2 bridges replication with other nuclear processes requires further support. The authors should provide direct evidence linking POLE2 to replication-coupled processes. The proposed POLE2 mutants could be used in replication studies to determine whether NPF binding influences DNA replication or the potential associated processes.

Version 1:

Reviewer comments:

Reviewer #1

(Remarks to the Author)

The authors have made a sincere effort to address the comments of all reviewers. They added additional data to enhance the biological implications of their findings, including POLE2 variants in vivo biotinylation experiments and immune fluorescence imaging data of the Y513C variant vs. wildtype. The authors have satisfactorily addressed all my comments/suggestions raised during the initial revisions.

A few noteworthy improvements: They reworked the entire last section (Fig.6), including updated supplementary Figures (Sup. Fig. S17 & S18), which are on point with their messaging. Several figure panels and figure captions (among them Fig. 1, Fig. 2, Fig. 3 and S1) were updated regarding labels, categories, color coding, which in my opinion largely increases readability. In addition, the authors increased data accessibility (deposited AF3 models on Zenodo, accessibility of single MS-raw files on MassIVE), ensuring reproducibility and transparency. They added a comment regarding limitations to their discussion sections, to address a comment from reviewer #3 regarding limitations of their in vitro affinity/hold up assays. It is important to note that the authors addressed the concerns/ and major concerns raised by the all three reviewers regarding the BiID experiments and reworked this section by including additional data and data representation. Overall, they used additional data to address the raised concerns of reviewer #2 and #3 regarding the biological implications of the NPF-mediated networks. I therefore recommend the manuscript for publication and look forward to seeing it in press.

Reviewer #2

(Remarks to the Author)

Generally the authors have satisfactorily addressed the comments that I made on the initial submission. The main strength of the manuscript is that it presents a thorough characterisation of NPF motif binding to POLE2. Further work is still required to assess the biological significance of novel POLE2 interactors, especially those not thought to be involved in DNA replication.

Reviewer #3

(Remarks to the Author)

While the manuscript has been improved, a key claim in the abstract remains overstated.

The statement, "Overall, these findings establish POLE2 as a central hub linking replication with other processes via broad NPF-motif recognition," is not fully supported by the data.

The study demonstrates POLE2's capability to interact with multiple NPF-containing factors, but it does not provide direct evidence that POLE2 functionally links replication to other processes.

To support this strong conclusion, functional experiments are needed; otherwise, this claim should be revised to only reflect the interaction data. Don't overstate.

REVIEWER COMMENTS

Reviewer #1 (Remarks to the Author):

The paper “The non-catalytic DNA polymerase subunit is an NPF motif recognition protein” focuses on systematic identification of proteins acting as receptors for the three amino acid long NPF-motif. Keskitalo and Zambo et al. utilized a combination of biochemical assays, a set of mutants and structural predictions to screen for NPF-motif recognition proteins. For this well characterized and understudied/ orphan NPF-motif containing peptides were used employing a quantitative MS-based hold up assay. They identified POLE2 as a reoccurring binding partner of multiple NPF-motifs. Affinity screens confirmed interactions of POLE2 with nuclear proteins containing NPF-motifs. PPI-database driven network analysis and a proximity interaction mapping approach were used to confirm the role POLE2 as a hub in the NPF-motif containing protein network.

SLiMs are difficult to study, as they are often in IDRs, typically only a few residues long, and lack structural folds. Therefore, most SLiMs currently listed in databases have been predicted using consensus sequence-based rules. To improve SLiM prediction algorithms, it is crucial to expand experimental data on SLiMs to reveal their role in protein-protein interactions. The Authors applied a combination of affinity-based assays (native Hold Up assays), sequence-based (motif) conservation predictions, mutational screens and structural predictions to identify NPF-motif receptors. This generates a useful resource that should be integrated into prediction algorithms of SLiMs, thus enhancing their performance.

The rationale of the experiments, the applied methods and the obtained results are mostly detailed and understandably described (see comments for a few suggestions). Overall, the work represents a significant effort to produce robust and comprehensive analyses of NPF-motif receptors. The figures describing this are mostly clear and convey useful and interesting insights. The authors have also been very up-front about the limitations of their employed methods/approaches (e.g.: Ex vivo holdup experiments are only capable to uncover the intrinsic properties of interaction networks, ...), which is appreciated and should be helpful to readers to judge the reported findings.

I congratulate the authors for their rigorous work on mapping novel NPF-motif receptors. I have only a few minor comments where I think a modest amount of rephrasing, clarification and data analysis could be helpful and I have no trouble recommending this manuscript for publication.

We greatly appreciate the reviewer’s supportive feedback on our work.

1. It is very much appreciated that raw/minimal processed western blot images (e.g.: Supplementary figures) are provided alongside the WBs and quantitative data shown in the main figures. Further, I want to acknowledge that the MS-raw data was deposited. However, the MS-data was deposited in a difficult to access way (controls and samples packed in *.zip folders with 200 – 300 GB). MassIVE supports the submission of individual raw files and promotes this data structure. Transferring a single packed raw file folder via FTP would have taken me several days, which is unacceptable and makes reusing the files tedious. Thus, I was not able to check the DIA-NN files completeness (raw files, parameter files, sample file list annotation, peptide or protein level outputs) nor the BioID files (raw files, MSFragger (FragPipe) parameter files, outputs). To facilitate easier access, please make the files available individually and provide sample annotation files.

Thank you for raising the issue of data accessibility. To make reuse straightforward, the repository contains two ways (same access code and password) to access the MS data:

1. Merged archives: the nHU analysis and matched control datasets are available as zipped bundles in the raw/ folder.
2. Per-run access: all individual RAW runs can be downloaded separately from the updates/ folder.

We apologize for the limitations of the repository/database platform, which encourages large compressed uploads and makes alternative structures cumbersome. Within these constraints, we have organized the data to support both complete one-shot downloads and selective per-run retrieval. We hope this resolves the accessibility concern.

2. I would encourage the authors to make the best performing models of the AlphaFold predicted structures (*.cif or *.pdb files) including the predictions of full-length complexes, accessible to the public by depositing the structures (e.g. Zenodo or a similar platform), including again annotation files.

The prediction files of full-length complexes, as well as the DONSON motif, together with all AF3 outputs, are now included as a supplemental dataset to the manuscript on the Zenodo platform with the following accession doi: <https://doi.org/10.5281/zenodo.16792938>

3. Introduction: “Two of these EH proteins contain three EH domains, four of them contains two domains and the four EHD proteins that contains a single EH domain form stable oligomers 14,15.” “contain” instead of “contains”. “Since all members of the EH family recognize the same structural motif, a putative NPF motif can interact with any members of the family.” “any member” instead of “any members”.

These grammar issues are now resolved.

4. “Identification of POLE/POLE2 as a recurring interaction partner of unrelated NPF motif containing peptides”; Section: “ ... unexpected to identify multiple types of evolutionarily unrelated NPF motifs as binding partners of same complex, including the well characterized...” “the same complex” or “same complexes” instead of “same complex”.

These grammar issues is now resolved.

5. “Structural insight into NPF motif binding by POLE2”; Section: To maintain consistency, please standardize the abbreviation throughout the document. Use either "AlphaFold 3 (AF3)" or "AlphaFold" consistently in all sections.

We standardized AF3 referencing.

6. “POLE2 interacts with NPF motifs proteome wide”; Section: “In this experiment, we performed an affinity measurement with 6516 endogenous, full-length proteins, covering 32 % of the human proteome.” There are 7,236 Protein Groups reported in Table S1, and 6,516 are matched with proteotypic peptides. Also, the Protein Groups which have multiple mapped UniProtKB identifiers are endogenous, full-length proteins. Rephrase and clarify that only proteins for which proteotypic peptides were found are included in the count of 6516 proteins.

We thank for the reviewer for pointing out this discrepancy. We rephrased as suggested and removed all ambiguous protein groups from the supplementary table that were anyway excluded

from the analysis.

7. “Validation of selected nuclear POLE2-NPF motif interactions”; Section: “Then, we cloned these three full-length proteins with an HA tag into a mammalian expression vector, produced cell extracts from transfected HEK293T cells and performed nHU titration experiments using purified His6-avi-MBP-tagged POLE2 as bait and analyzed the results with dot blot (Figure 4B, S13, S14, S8B, Table S2).” The result section discusses three selected proteins (TTF2, ERCC6L and WDHD1), but Supplementary Figure S14 references two additional proteins, PYGO2 and CLSPN which were not discussed in this section of the manuscript. Is this link wrongly set or was the text adapted to three candidates which worked? This is confusing, especially as PYGO2 and CLSPN were not significant in the POLE2 nHU-MS experiment shown in Fig. 3A (see also comments for Figure 3A).

We did perform additional experiments with two other HA-tagged proteins, namely SSBP2 and AGFG1. While their expression was initially confirmed in Supplementary Figure S6C, the corresponding dot blot experiments did not yield successful results, as shown in Supplementary Figure S13. In light of this, we did not pursue further follow-up experiments with these proteins or their NPF motif mutant versions. Nonetheless, we have included these negative results in the manuscript, both for the sake of transparency and to ensure the raw dot blot data are represented without omission.

Supplementary Figure S14 originally contained results from nHU titrations performed between HA-tagged proteins and POLE2 baits, as well as with endogenous POLE and NPF motif peptide baits. These were not easily incorporated into the main manuscript without disrupting clarity and flow. In this figure, the lower panels correspond to dot blot experiments with HA-tagged proteins, whereas the upper panels, clearly labeled in both the axes and figure legends, correspond to titrations of PYGO2 and CLSPN NPF motif peptides against endogenous POLE. The raw data for these measurements are also provided in Supplementary Figure S12.

We fully appreciate that this method of data grouping could lead to misinterpretation. To prevent this, we have now moved the peptide titrations with endogenous POLE from Supplementary Figure S14 to Supplementary Figure S12, and we have also revised the main text to refer to them more explicitly and clearly.

8. “In vivo proximity interactome of POLE2”: Section: “By analyzing the cellular localization of the identified partners, we found that all of them are located either in the nucleoplasm, or in the centrosome, indicating a nuclear and a cytoplasmic POLE2 cellular pool (Figure 6B).” Was GSEA performed (GO CC) or were the localizations manually curated (in Figure 6B, the label for the legend is missing). What thresholds for the GSEA were considered to draw the conclusion regarding localizations? Please clarify.

The corresponding section of the manuscript was substantially rewritten and the new version of the manuscript does not contain GSEA analysis. In the new version, the partners of POLE2 variants were manually clustered based on GO annotations.

9. “In vivo proximity interactome of POLE2”: Section: “We also find a particularly strong enrichment of proteins involved in biological processes related to DNA replication, DNA damage response, and cell cycle control, processes in which POLE has known functions (Figure 6C).” What GSEA test was used? What gene sets/ terms were used (GO BP?). What thresholds were used? What does “strong” enrichment mean?

The new version of the manuscript does not contain such analysis anymore.

10. General comment regarding GSEA/ GO enrichment: GSEA is not covered in the methods. Please consider adding a brief paragraph to the method section.

We agree that this section was poorly described in the previous version. However, we realized during the revision process that this section did not provide very useful insight to the manuscript and decided to completely remove it.

11. Discussion: “Out of these, POLA1 is an interaction partner of WDHD1 and may be captured indirectly.” Lysis conditions for BioID experiments (0.1% SDS) were not native. BioID captures direct or indirect proximity interaction partners, as the enrichment of interactors depends only on the distance to the tagged bait. In BioID direct or indirect can only be inferred using database or literature knowledge.

We fully agree with the reviewer that BioID, due to its lysis conditions, does not distinguish between direct and indirect partners that are not biotinylated themselves but are instead captured via a biotinylated interactor. At the same time, indirect partners are still within the proximity proteome of the tagged protein and therefore are likely to undergo biotinylation as well. In this way, BioID experiments capture both direct and indirect interaction partners.

As the reviewer correctly notes, the possible directionality of interactions can only be inferred from existing literature or database knowledge. This is precisely what we aimed to do in the sentence in question. The interaction between WDHD1 and POLA1 has been previously reported, and thus it is reasonable to speculate that one may associate with POLE2 through the other. At this point in the manuscript, the interaction between POLE2 and WDHD1 was already established, whereas the interaction between POLE2 and POLA1 was not. For this reason, we described POLA1 as being “captured indirectly.” However, as we clarify in the following sentences, POLA1 also contains an NPF motif capable of direct binding to POLE2. Thus, our final conclusion is that both WDHD1 and POLA1 can, in fact, represent direct partners.

To avoid ambiguity, we have revised this sentence for clarity and included an appropriate reference to support the interaction.

12. Methods: “Single-point and titration nHU experiments”:

Was Benzonase/DNase treatment performed on Yurkat cell extracts? Benzonase treatment was performed for the BioID experiment in Hek293 cells. If no Benzonase was added, can you exclude co-depletion of proteins through DNA stretches? In the manuscript you report that POLE2 was most likely depleted through its interaction with POLE. How can you rule out that other nuclear/DNA binding proteins were not only depleted through their binding to a DNA-strand on which a NPF-motive receptor protein is too binding?

We thank the reviewer for raising this important point. No DNase treatment was included during extract preparation for the nHU experiments, as this step would considerably reduce the proteomic depth of the mass spectrometric analysis. While this is not a concern for BioID experiments due to the strong enrichment and washing steps, holdup does not involve washing, and any added DNase would remain present during proteomic measurements. Instead, to address DNA content, the cell extracts were aggressively sonicated, as described in the Methods section, to fragment DNA extensively.

We acknowledge that this approach cannot completely exclude the possibility that some interactions may be mediated through DNA stretches. However, we believe this is unlikely for several reasons. To detect a measurable depletion in holdup experiments, at least 10–20% of partners must be captured. Achieving this would require resin-immobilized baits to be present in stoichiometric or even superstoichiometric amounts relative to the interacting proteins, depending on binding affinity. For indirect capture through DNA to occur, our resin would therefore need to be extensively coated with DNA.

If this were the case, we would expect to observe a substantial enrichment of general nucleic acid-binding proteins, such as histones, similar to the interactomes obtained when using small DNA baits in nHU experiments (see Zambo et al., 2024, BioRxiv). In that recent study, using ~20 bp dsDNA baits, we identified hundreds of partners. By contrast, for POLE2 we detect far fewer partners, with minimal overlap with typical dsDNA-binding proteins. This strongly argues against the presence of substantial DNA contamination on the resin.

That said, we cannot entirely exclude that trace amounts of DNA may remain. However, we are confident that any such minor DNA contamination would not be sufficient to cause partner depletion within the detection limits of our experimental setup.

13. Methods: “Statistical and bioinformatics analysis of the proteomics data for nHU samples”: “The input file to further dia data analysis was the DIA-NN Report.pg_matrix.” Rephrase (“to” -> “for”)?

We rephrased as suggested.

14. Methods: “AlphaFold predictions”: “In all instances when this pocket was utilized for interactions, the pLDDT scores of the docked NPF or NPY motifs were outstanding compared to their environment and were similar to the scores of the globular POLE2 regions.” Either rephrase or provide a comparison (e.g. pLDDT scores of the NPF and NPY motif versus the rest of the protein/peptide sequence). Outstanding cannot be judged with the information provided in the manuscript/and or methods. Another option would be to mutate the NPF or NPY motifs (e.g. to KAF), perform AF3 predictions and check for significant (“outstanding”) difference in the scores vs. the peptides with NPF or NPY motifs. Further, the pLDDT is a measure for the confidence in the local structure for each residue. Please also provide the ipTM scores, which is a measure for the relative position of the two chains. A table with all AlphaFold models and scores would be useful to judge the significance of the finding.

We have rephrased the sentence in the manuscript to improve clarity.

A visual comparison of pLDDT scores was already included in Supplementary Figure S11, which shows the identified NPF motif sequences from all AF3 predictions, color-coded according to pLDDT scores. In these images, the distinction is visible: the core NPF motifs consistently display higher scores than their N- or C-terminal extensions. This is reflected by the coloring, where the outer peptide regions (shown in cartoon and stick representations) shift toward the red end of the spectrum (lower scores), while the core NPF tripeptides are closer to the blue end (higher scores). We believe this provides a clear visual demonstration that the core NPF motifs are predicted with higher confidence than their surrounding regions.

With respect to mutational testing in AlphaFold, while technically feasible, such analyses would be of limited interpretive value. By removing the NPF motif, the corresponding binding interface would no longer exist, and the outcome would largely confirm this expected loss of interaction. We feel these kinds of assessments are more informative when performed experimentally, as we have

done in this study across multiple targets, where they provide direct evidence for NPF motif-mediated binding.

In addition, although we consider pLDDT scores the most relevant metric here—since they reflect local confidence in short motif-mediated interactions—we have now also included ipTM score analyses in Supplementary Figure S11. All AlphaFold model outputs have been deposited in the Zenodo supplement so that readers may perform any additional analyses according to their preferences. The ipTM score analysis revealed that partners containing NPF motifs scored among the highest, while some partners interacting through other POLE2 interfaces also achieved relatively strong scores.

We hope these revisions and additional data presentation will address the reviewer's concern and make our reasoning clearer to the reader.

15. Methods: BioID experiments: "... and 50 μ M biotin added to 3 h before the harvesting to induce the biotinylation .." added to the media missing?

We rephrased as suggested.

16. Methods: "Identification of High-Confidence Protein-Protein Interactions of BioID experiments": "Interactions that passed the filter with any bait were rescued for other baits as well." Were any additional baits used for BioID besides POLE2? If not, please rephrase or remove the statement. If there were additional baits use, they were not reported in Table S4.

This was resolved in the revision.

17. Methods: "Identification of High-Confidence Protein-Protein Interactions of BioID experiments": "High-confidence interactions (HCIs) were defined by an estimated protein-level Bayesian FDR (BFDR) of ≤ 0.01 ." Table S4 reports 48 are proximity interactions. In Fig. 6A only 24 are shown (see also Fig.6 comments). It seems like there are additional 22 PPIs with a BFDR larger 0.01 and smaller or equal to 0.05 reported. Maybe the wrong version of the table was uploaded? Either remove the additional proximity interactors or discuss them in the manuscript and adapt the filtering threshold. To ensure full transparency it would be appropriate to report the full unfiltered scoring output containing background preys and scored proximity interactors.

This was resolved in the revision.

18. Figure and Figure legends: Figure 1A: "multiple types of receptors" – the proteins shown are according to your manuscript and UniProt not classical receptors. Please ensure that "NPF-motif receptors" is specified in all instances where "receptor" is mentioned in the manuscript, as done in the figure caption of Fig. 1C-H.

In biochemical contexts, the term *receptor* is sometimes used broadly to describe a molecular site that specifically recognizes and binds another molecule. In our manuscript, we use the phrase "NPF motif receptors" to denote proteins that specifically bind peptide segments containing NPF sequences. We chose this terminology to highlight their role as recognition modules for this motif, in analogy to how other binding proteins are occasionally described as receptors in structural and biochemical literature. This terminology also helps us discuss different, otherwise unrelated protein groups that all bind to the same ligand in a concise and consistent way.

We thank the reviewer for pointing out that in a few instances the term “receptor” was used without the full qualifier, which could potentially cause misunderstanding. We have revised the text to consistently refer to “NPF motif receptors” throughout the manuscript in order to avoid ambiguity.

19. Figure and Figure legends: Fig.1C-H. I struggle to comprehend why non-significant proteins are highlighted in the same manner as significant NPF-motif recognizing proteins. For example, in Fig.1C, ITSN1 (not significant) is highlighted and labeled, close to it is another highlighted point but not labelled. I suggest using a different label/color, or ring not filled out for non-significant but still discussed examples, instead of a full point. State statistical thresholds (\log_2FC and p-value) in the figure legend and/or main manuscript (section “Identification of POLE/POLE2 as a reoccurring interaction partner of unrelated NPF motif containing proteins”). Highlighted points which have close to no change, for example for WEE1_228-447 (Fig. 1F), BORA_18-38 (Fig.1G) should be differentiated to significant hits.

The inclusion of non-significant EH and POLE/POLE2 proteins in the figures was intentional, as it helps to demonstrate the specificity of NPF motifs. Motifs are sometimes assumed to interact broadly with an entire family of recognition domains (for example, that an NPF motif is simply “a binding motif of EH domains”). Our data show that this is not the case, since not all members of the same EH family bind equally to the studied NPF motifs. Similarly, in the case of POLE/POLE2, the fact that binding is observed for only a subset of motifs argues against this protein being either a general NPF motif binder or a common background protein frequently depleted in nHU experiments, as sometimes encountered in “crapome”-type datasets.

Regarding the reviewer’s note on the labeling next to ITSN1: this point does correspond to EHD4, but we recognize the arrow placement could cause confusion. We have now adjusted the figure to make this clearer.

We also acknowledge the reviewer’s point that significance thresholds can be subjective in such analyses. In our study we chose to apply a more stringent thresholding approach than is commonly used in the field. While more conventional approaches would identify additional partners, we prefer a conservative strategy to minimize false positives. For this reason, we did not highlight proteins that fall just below our chosen threshold, as this could raise unnecessary concerns. However, we have been transparent in the text that such partners are indeed below our applied thresholds, leaving it to the reader’s judgment to interpret these data further.

To aid clarity, we have added a new section in the Methods describing our thresholding approach in more detail.

20. Figure and Figure legends: Figure 2: “AlphaFold predictions confidently dock NPF peptides onto a solvent-exposed surface of” Provide the ipTM-score to support your statement (see also method statement for structure predictions with AlphaFold).

We included the ipTM score on Figure 2. In addition, the new supplement contains all predictions so the readers have the possibilities to perform additional analysis on these simulations.

21. Figure and Figure legends: Figure 3: A) “Those significantly depleted partners that contain putative NPF motifs are highlighted with green, or blue circles.” Why is PYGO2 highlighted? It does not seem to be significant (\log_2FC -0.63 and $-\log_{10}(p\text{-value})$ of 1.36). Was it highlighted because it contains the NPF motif? If so, rephrase the figure caption for clarity. Further, you state in

the section “Intrinsic connection between distinct NPF motif-mediated networks” that there is no interaction. This needs to be revised.

PYGO2 was included in the plot to support the following statement: “POLE2 was also found to interact with the WNT-enhanceosome and with many of their known partners, such as SSBP3, LMO4, LEF1, but not with PYGO2”. This is now further clarified in the legend.

PYGO2 is highlighted because multiple components of the WNT enhanceosome, the canonical binding partner of PYGO2, was detected as a POLE2 binding partner. In fact, SSBP2 and other components of the complex are even highlighted with a distinctive color on Figure 3A and Figure S9 to keep consistency with the coloring in Figure 1. The reason to show that PYGO2 is not binding to POLE2 is showing that the WNT enhanceosome can bind to POLE2 in the PYGO2-free state.

22. Figure and Figure legends: Figure 3: The upper panel is challenging to read. How many putative motifs of partners were overlaid? Please provide the number (N). Are these the 7 motifs mentioned in the main manuscript text and shown individually in Supplementary Figure S.11? I advise showing only one individual structure. Consider pushing the overlay to the supplementary figure instead. Please provide AF scores (e.g. pIDTT, ipTM, pTM) in a supplementary table or add them to the figure caption.

The figure legend was refined to clarify the issue. All predicted 7 motifs were overlaid and the goal of the panel is to show that although the extensions of these motifs do not overlap, hence “challenging to read”, the core NPF sequences perfectly align. On a single structure, such information would simply disappear. To get more atomic detail of NPF interactions, one can use the lower panel, or Figure 2 where the NPF motif binding of POLE2 is shown in detail, or Figure S11 where the structures shown individually.

On Figure S11, ipTM scores of each prediction is now shown. All predicted structures, alongside various metrics are now included in the Zenodo supplement, allowing readers to perform more analysis, according to their preferences.

23. Figure and Figure legends: Figure 6A: “(A) Identified partners of POLE2, ranked by their enrichment.” The plot shows average signal (Average Spectral Counts) and not enrichment against GFP controls. Rephrase statement or change labels. In the main manuscript 26 proximity interactors are mentioned (...the BioID experiment revealed 26 significantly enriched partners ...). In the panel only 25 are shown, and comparison with Table S4 shows that TUBB4A was removed. Why? Add the number of replicates and the number of GFP-controls used to the figure legend.

This was resolved in the revised manuscript.

24. Figure and Figure legends: Figure 6C: To improve clarity, please add a color key for the different node coloring used to the legend. Additionally, the varying edge thickness should be explained in the legend of the plot, indicating what it represents. Also, specify if a significant threshold was used to filter GO terms in the figure legend.

This was resolved in the revised manuscript.

25. Supplementary Figures: Fig. S1B: provide reference to database form where the domain architecture was obtained (InterPro, UniProtKB).

The source of the domain architectures is now clearly defined.

26. Supplementary Figures: Fig. S2B: What is the number of replicates (n=2 biological replicates for each of the 2 blots shown in Figure S3?) for the bar plots shown in panel B? Does the plot show mean, and standard deviation? Clarify.

These pieces of information are now included in the legend.

27. Supplementary Figures: Fig. S9: Explain the color code that the figure is readable (same color code as in Figure 3?).

The color coding is clarified in the legend.

Reviewer #2 (Remarks to the Author):

This manuscript by Keskitalo et al investigates the role of the NPF SLiM motif in mediating protein protein interactions. They use an approach that they developed called native holdup to screen for binding partners of several different NPF protein including two factors involved in DNA replication (DONSON and WDHD1). One of the top hits from the screens is a subunit of DNA polymerase epsilon called POLE2. A second screen is performed using POLE2 as bait that identifies a number of NPF containing proteins. Mutational analysis based on alphafold 3 predictions convincingly locates the NPF binding site on POLE2. BioID with POLE2 is then used as an alternative method to assay putative POLE2 binding partners. However, this approach only identifies 3 proteins that were also found by nHU. It is then shown that POLA1, which is seen only in BioID, binds to POLE2 also via an NPF motif interaction that is predicted by alphafold.

This is a nice study that helps to define POLE2 as an NPF binding protein. The use of an unbiased approach to identify binding partners of POLE2 demonstrates that POLE2 can bind to diverse NPF containing proteins, many of which are not thought to be involved in DNA replication. However, most of these proteins were not detected by BioID, casting doubt on whether their binding to POLE2 has any biological relevance. The manuscript clearly shows that DONSON, TTF2, WDHD1 and POLA1 can all bind to POLE2 via an NPF-mediated interaction. This has already been shown in published work for DONSON and in recent pre-prints for TTF2. That being said, this manuscript nicely illustrates that there is likely to be competition at replication forks for a single NPF binding site on POLE2. Overall, the experiments are well performed and the data are appropriately presented. Due to a lack of any new functional data the findings presented are a little limited because it remains unclear if the binding to POLE2 to diverse NPFs (beyond known DNA replication factors) has any function. New insights in this direction would enhance the impact of the work. Some specific comments to consider are listed below:

We thank the reviewer for the thoughtful evaluation of our study. With respect to the concern about the limited overlap between the different interactomic approaches, we would like to clarify that BioID, which captures the proximity proteome of POLE2 at a given cellular stage, is strongly condition dependent. By contrast, the affinity interactome of POLE2 measured by nHU reflects intrinsic binding affinities, which are almost always not condition dependent. This distinction explains much of the discrepancy observed between the two approaches. For example, it is likely that additional partners would be detected in BioID experiments if the cells were subjected to specific signaling events or DNA damage, whereas the intrinsic affinities would remain unchanged, as they are thermodynamic constants. This is illustrated by TTF2, an established POLE2 partner that was identified only in the affinity screening.

Regarding the lack of functional data, we have now strengthened the manuscript by including a new series of experiments examining the effects of mild missense point mutants on the cellular

localization and proximity interactome of POLE2. These results provide functional evidence supporting the cellular importance of the interaction network we describe.

Specific comments

Figure 1 – it might be nice to see peptide sequences. What was the rationale for the sequence choice? e.g. total length, number of residues either side of the NPF?

Peptide sequences are now included in the method section.

Each peptide was designed to take into consideration their evolutionary conservation pattern, as well as practical aspects, such as predicted solubility of the resulting synthetic peptide. Moreover, we tried to avoid placing the core NPF motif close to either ends of the peptides.

The authors claim that it is surprising that the NPF motifs of DONSON and WDHD1 bound to POLE2. Given that the Walter lab showed that DONSON binds to POLE2 via this motif in a manuscript published in 2023, and the WDHD1 POLE2 interaction is one of the top hits in the predictomes.org database for WDHD1, the result does not seem particularly surprising.

We appreciate the reviewer's perspective that these interactions may not be surprising in light of prior data. At the same time, we note that before our study it was not possible to assess the specificity of these interactions, and in the case of WDHD1, even the validity of the earlier prediction remained uncertain.

During the course of our investigation, the interaction between DONSON and POLE2 was indeed reported by others, and we agree that the WDHD1–POLE2 interaction ranks highly in the Predictome predictions. However, to our knowledge, no experimental evidence for this interaction was available prior to our work. It is also worth noting that POLE2 has many other high-scoring predicted interactions in the Predictome database that we were unable to validate experimentally in either of our interactomic screens. This highlights that large-scale AlphaFold-based prediction efforts, while valuable, also carry a considerable false-positive rate when experimental support is lacking. In our view, this underscores the importance of combining predictive approaches with systematic experimental validation, as such predictions can indeed point to genuine interactions but must be interpreted with caution.

We therefore see Predictome and similar resources at the current state of science as highly useful hypothesis-generating tools, whose greatest value lies in guiding experimental work rather than serving as stand-alone evidence.

The measurements of the DONSON NPF binding to POLE2 are quite variable across different experimental set ups, as the authors acknowledge. Therefore, because binding of this region of DONSON to POLE2 has previously been demonstrated, the experiments don't add much new insight.

We would like to reassure the reviewer that the measured affinities do not vary to an extent that would raise concern. The dissociation constant of HA-tagged POLE2 was consistently observed in the range of ~300–500 nM for the DONSON motif, while MBP-tagged POLE2 bound more weakly, with dissociation constants in the range of 1–10 μ M depending on the experimental setup - a difference that is most likely explained by the well-documented tendency of the bulky MBP tag to perturb protein interactions.

Importantly, these results were always internally consistent within each assay. Furthermore, even if one takes a cautious approach and places less emphasis on the absolute values, the key experiments shown in Figure 2 can also be interpreted based on relative affinities, without altering the central conclusions of the study.

We would also like to highlight that, prior to our work, no biophysical characterization of this interaction had been carried out, and the POLE2 interface itself had not been experimentally validated beyond AlphaFold predictions. In this study, we performed a comprehensive biophysical characterization at multiple levels (in cells, ex vivo, and in vitro) and carefully validated the predicted interface. We therefore believe that our work provides substantially deeper insight into this interaction than what was previously available.

The AlphaFold prediction of the DONSON POLE2 interaction is described in some detail, perhaps too much detail given that it is based on an in silico prediction and not an experimentally derived structure.

We agree that such predictions have to be analyzed with strong skepticism, however our findings are later thoroughly validated experimentally and in order to interpret those results and their rationality, it is key to analyze the predicted structures in details.

The results of experiments with POLE2 variants are interesting. It is concluded that the Y513C variant abolishes NPF binding. How frequently does this variant occur and is it associated with any phenotypes? Generally, a bit more background here would be helpful. Loss of binding of POLE2 to NPF factors could interfere with multiple important replication processes that could each have phenotypic consequences.

According to our data, both natural POLE2 variants impacts NPF binding; Y513C leads to full and D284G to partial loss of function in ex vivo nHU experiments and in cells, many interactions are similarly strongly impacted by the D284G mutation as with Y513C. We also show substantial evidence in the revised manuscript that the Y513C mutation causes decreased stability and increases the tendency of POLE2 to form large cytosolic aggregates. These findings raise the clinical importance of these natural POLE2 variants. Fortunately, both are rare in the population, but both were observed on multiple occasions. In the gnomAD database, the Y513C variant was identified in 5 instances out of 1572996 measured alleles and the D284G was found in 4 alleles out of 1610798 measured ones.

When TTF2 is introduced, the description should be more precise and the relevant pre-prints cited. I think the work from the Labib and Walter / Pellman labs clearly shows that TTF2 is needed for mitotic replisome disassembly and that this requires the NPF-mediated interaction of TTF2 with POLE2. The description in the discussion is more appropriate and could be moved forward.

We agree with the reviewer that the recent preprints link TTF2 to mitotic replisome disassembly, and we had already acknowledged this in the discussion. At the same time, we note that this is not necessarily considered the major or most widely accepted function of TTF2, nor does it represent the only possible mechanism by which its interaction with POLE2 may be relevant. The studies from the Labib and Walter/Pellman groups provide compelling evidence for an NPF-mediated role in replisome disassembly; however, they do not explore whether this interaction might also contribute to other cellular processes.

We believe that further investigation will be important to clarify whether POLE2–TTF2 interactions may also play a role in transcription regulation or in other contexts. To reflect the reviewer’s suggestion, we have now expanded the discussion of these preprints in the section where TTF2 is introduced, so that their findings are presented more clearly.

Are any of the EH binding proteins that were found to bind POLE2 seen in any of the iPOND data sets? This analysis would be helpful because it might provide some insight into whether any of these proteins are ever found at or near to active replication forks.

We kindly ask the reviewer to specify which iPOND resource they had in mind, as PubMed lists more than 50 studies using this approach. From the datasets we have examined, POLE itself appears to be only weakly enriched in iPOND analyses (for example, see Figure 4 of Sirbu et al., JBC 2013), making it difficult to draw strong conclusions regarding this protein from such data.

The authors say that Exo1 was identified in nHU and BioID as a POLE2 binding partner. However, Exo1 does not appear in the list of proteins shown in Figure 3 and there doesn’t seem to be any further discussion about Exo1.

It is correct that while Exo1 was identified as a confident binding partner of POLE2, it does not appear in the table of identified NPF motifs in Figure 3. The reason is that Exo1 does not contain a recognizable NPF motif, and we therefore suspect that it binds at a different site on POLE2, which will be interesting to explore in a future study.

The case of Exo1 became even more intriguing in the light of the new experimental data with the revision suggesting that Exo1 binds via the NPF binding interface. To resolve this discrepancy, we started further investigations and found that although our AF3 predictions did not reveal a meaningful interface between these proteins, AF2 predictions available in the Predictome database suggested that Exo1 may interact with the same POLE2 pocket via an NKF motif (residues 481–483). Since the local pLDDT scores for this prediction are relatively low and this interface could not be reproduced with AF3, we remain cautious about the plausibility of this interaction mode and it remains a possibility that Exo1 interacts with POLE2 indirectly.

Overall, we consider Exo1 a promising candidate for further investigation, and we look forward to addressing its binding mode in future studies.

The authors assume that the interaction of POLE2 and WDHD1 is “of critical cellular importance”. However, they provide no data pertaining to the function of this interaction in any aspect of DNA replication and therefore it still needs to be established if the interaction is critically important.

It is correct that our study does not attempt to functionally characterize the identified interactions in detail. However, we do show that on a proteome-wide scale WDHD1 emerges as the most significant interaction partner of POLE2 and also as the highest-affinity NPF motif-containing partner (see Figures 3 and 6). The NPF motif of WDHD1 was originally identified using the SlimPrints approach, which detects highly conserved motifs across proteins. This analysis highlights that the motif is exceptionally well conserved, as exemplified by its presence in the WDHD1 homologs of very distantly related organisms, such as the spreading-leaved earth moss *Physcomitrium patens* (UniProt ID A0A2K11D13) and the marine diatom *Thalassiosira pseudonana* (UniProt ID B8C836).

This degree of conservation is notable, since short linear motifs typically evolve rapidly and can even appear or disappear among closely related species. Only the most functionally critical motifs tend to be preserved across multiple kingdoms. In addition, recent structural studies of the replisome revealed the close spatial proximity of WDHD1 and POLE2 in active complexes, even capturing this interaction, as we also discuss in our manuscript.

Taken together, these findings strongly support the functional importance of the WDHD1–POLE2 interaction. While further studies will be needed to explore its precise role, we believe the evidence already provides compelling grounds to view this interaction as biologically significant. We thank the reviewer for highlighting the importance of functional characterization, which we agree is an essential next step.

This work establishes nicely that at least 5 replication proteins (WDHD1, EXO1, TTF2, POLA1, DONSON) bind to the same site on POLE2. This suggests a competition between these factors for binding which is an interesting point for discussion. For example, how does TTF2 bind for replisome disassembly when presumably WDHD1 and POLA1 are still present at forks? Is binding regulated, perhaps via PTMs?

We were not able to find clear evidence for post-translational modification regulating these interactions. Since the interactions are mutually exclusive, some form of competition must exist *in vivo*. One way to reduce this complexity is to consider that replisome composition is itself dynamically controlled, such that only one partner may be present in the proximity of POLE2 at a given time. For example, during assembly, DONSON may represent the dominant partner, whereas during disassembly, TTF2 could take on the key role in the absence of other factors. Likewise, during active replication, WDHD1 and POLA1 may be the primary interacting partners.

From a biochemical perspective, however, it is perhaps more likely that many of the interactions we identified—more than a dozen in total—can occur simultaneously, but in a highly transient and dynamic equilibrium. This behavior would be consistent with the principles of the law of mass action and related physical laws that govern binding equilibria.

We thank the reviewer for raising this important point, which we believe highlights an interesting direction for future investigation.

Reviewer #3 (Remarks to the Author):

In this study, Keskitalo et al demonstrate that POLE2, the non-catalytic subunit of human DNA polymerase epsilon, acts as a receptor for NPF-motif-containing peptides. The authors identify multiple NPF-containing peptides and nuclear proteins that interact with POLE2, validating some interactions through biochemical assays and BioID experiments. Based on these findings, they propose that POLE2 serves as a central hub connecting DNA replication with other nuclear processes via its NPF-binding pocket. While the study presents novel and intriguing observations regarding POLE2's role in nucleating NPF-containing factors, the evidence supporting a direct functional link between POLE2-mediated NPF interactions and DNA replication remains incomplete. The mechanistic and functional insights are currently limited. With additional experimental validation, the manuscript could be strengthened. In its present form, it may be better suited for a specialized journal focusing on protein-protein interactions or DNA replication machinery.

We thank the reviewer for this valuable feedback and for highlighting the importance of functional relevance. We respectfully note that, at the molecular level, a physical interaction can itself be considered a functional property. By revealing the NPF-binding capacity of POLE2, we therefore provide meaningful functional insight into the protein's behavior. Likewise, by examining the specificity determinants that distinguish POLE2 from other NPF-binding receptors, such as EH proteins, we contribute to the functional characterization of the broader cellular NPF-motif interaction network.

We understand, however, that the reviewer is referring to the biological consequences of the identified interactions. This raises two key points. First, given the large number of interactions we have uncovered, a full functional characterization of each would require several years of work and is beyond the scope of the present study. Second, if we were to focus on only a single interaction, this would shift the emphasis of the manuscript away from the overarching network of NPF motifs toward one specific case, which was not our intention.

To address this concern, we performed an additional series of experiments using POLE2 mutants that alter NPF binding. Based on our structural validation (Figure 2), we selected both natural and artificial POLE2 variants and assessed their cellular localization and proximity interactomes. These experiments revealed that even mild missense point mutations can affect NPF-mediated interactions in living cells, and one natural variant even led to aberrant protein localization. In this way, we demonstrate that NPF motif binding has a clear functional consequence for the cellular properties of POLE2.

We thank the reviewer again for encouraging us to expand this aspect of our study, which we believe has strengthened the manuscript.

Major Concerns:

1. While the interactions are well-documented, the functional relevance of POLE2's NPF-binding activity remains unclear. Additional experiments would help establish whether these interactions are biologically meaningful.

The major goal of our study is to comprehensively map the interactome of POLE2 mediated by NPF motifs, a focus that is critical for understanding specificity within this interaction network and to establish key relationships with other, better studied NPF-mediated networks. We agree with the reviewer that determining the precise functional consequences of individual NPF-mediated interactions would be valuable, however, characterizing each interaction in depth would represent a substantial work and would shift the scope away from our primary aim of mapping the global interactome. To show the importance of these interactions, we carried out more extensive proximity labeling experiments, complemented with microscopic imaging on POLE2 variants to demonstrate how the collective loss of NPF motif interactions globally reshape the behavior and environment of cellular POLE2.

2. To better assess the importance of NPF binding, the authors should generate POLE2 mutants defective in NPF-motif recognition and analyze their impact using mass spectrometry to identify lost interactions.

We are glad for this proposed experiment by the reviewer that we included in the revised manuscript. This analysis revealed multiple interactions perturbed by these interface mutations including some that were already characterized and some that can initiate exciting future research directions. This analysis also revealed that certain interface mutations significantly impact POLE2

stability. In support of this, altered cellular localization was similarly observed for pathogenic POLE2 variants. This is significant especially because we studied very conservative mutations, such as the solvent exposed S522A that would typically not raise such concerns. Although it is speculation and cannot be tested easily, this phenomena can be explained with a mechanism that involves NPF motif-mediated interactions during the maturation of the epsilon polymerase complex.

3. The claim that POLE2 bridges replication with other nuclear processes requires further support. The authors should provide direct evidence linking POLE2 to replication-coupled processes. The proposed POLE2 mutants could be used in replication studies to determine whether NPF binding influences DNA replication or the potential associated processes.

The role of POLE in various nuclear processes beyond replication is already well established. Although POLE2 is mentioned less frequently in this context—likely due in part to the lack of suitable antibodies—it forms a stable heterodimer with POLE, and thus it is reasonable to conclude that POLE2 is also present and most likely actively involved in these processes. In our study, we link POLE2 to many proteins associated with nuclear functions outside replication, including several that resemble mechanisms in which POLE has already been implicated, such as DNA damage repair.

That said, we agree with the reviewer that each of these newly identified interactions, which suggest broader roles for POLE/POLE2, warrants further detailed investigation in future studies.